# Exposure to authoritarian values leads to lower positive affect, higher negative affect, and higher meaning in life

**Jake Womick**[1]*, **John Eckelkamp**[2], **Sam Luzzo**[2], **Sarah J. Ward**[3], **S. Glenn Baker**[4], **Alison Salamun**[2], **Laura A. King**[2]

**1** University of North Carolina, Chapel Hill, North Carolina, United States of America, **2** University of Missouri, Columbia, New York, United States of America, **3** University of Illinois, Urbana-Champaign, Illinois, United States of America, **4** Reed College, Portland, Oregon, United States of America

* jjwzp5@mail.missouri.edu

## Abstract

Five studies tested the effect of exposure to authoritarian values on positive affect (PA), negative affect (NA), and meaning in life (MIL). Study 1 ($N = 1,053$) showed that simply completing a measure of right-wing authoritarianism (vs. not) prior to rating MIL led to higher MIL. Preregistered Study 2 ($N = 1,904$) showed that reading speeches by real-world authoritarians (e.g., Adolf Hitler) led to lower PA, higher NA, and higher MIL than a control passage. In preregistered Studies 3 ($N = 1,573$) and 4 ($N = 1,512$), Americans read authoritarian, egalitarian, or control messages and rated mood, MIL, and evaluated the passages. Both studies showed that egalitarian messages led to better mood and authoritarian messages led to higher MIL. Study 5 ($N = 148$) directly replicated these results with Canadians. Aggregating across studies ($N = 3,401$), moderational analyses showed that meaning in life, post manipulation, was associated with more favorable evaluations of the authoritarian passage. In addition, PA was a stronger predictor of MIL in the egalitarian and control conditions than in the authoritarian condition. Further results showed no evidence that negative mood (or disagreement) spurred the boost in MIL. Implications and future directions are discussed.

## Introduction

Right-wing authoritarianism involves submission to a strong leader, endorsement of hierarchical social structures, strict conformity to social conventions and traditions, and aggression towards outgroups perceived as violating these conventions [1–3]. Right-wing authoritarianism involves anti-democratic and anti-social impulses, such as prejudice [1,2,4–6]. First studied by psychologists to understand the rise of the Nazis during World War II [1], right-wing authoritarianism remains relevant in contemporary scholarly research [7] and everyday life [8–10]. In a 2011 World Values Survey of 34 countries in the OECD, 44 percent of non-college graduates endorsed the authoritarian idea of "a strong leader who doesn't have to bother with congress or elections" [11]. Right-wing authoritarianism (particularly authoritarian aggression) played a significant role in the 2016 U.S. presidential election [7,12,13]. In subsequent

**Funding:** The author(s) received no specific funding for this work.

**Competing interests:** The authors have declared that no competing interests exist.

years, there have been numerous "alt-right" demonstrations in the U.S., including the 2017 "Unite the Right" rally, that culminated in a fatal car attack [14], and the 2021 Capitol Insurrection. In the U.S., between 2016 and 2017 the number of attacks by right-wing organizations quadrupled, outnumbering attacks by Islamic extremist groups [15], constituting 66% of all attacks and plots in the U.S. in 2019, and over 90% in 2020 [16]. Understanding the appeal of authoritarian values is a challenge for contemporary society and an important goal for psychological science.

What explains the appeal of authoritarian values? What problem do these values solve for the people who embrace them? Focusing on the North American cultural context, our approach to these questions assumed that attachment to these values must originate somewhere: The presentation of authoritarian values must have a positive influence on something that is valuable to people. Here we consider two experiences that might help explain the appeal of such messages, mood and meaning in life. We expected authoritarian messages to have divergent effects on these outcomes, worsening mood but enhancing meaning in life. We propose that authoritarian messages influence people on two separable levels, the affective level (lowering positive and enhancing negative affect) and the existential level (enhancing meaning in life). Understanding the appeal of authoritarianism requires attention to both levels of psychological functioning. Before presenting the studies, we define the outcomes of interest and then consider their relationships to right-wing authoritarianism.

## Meaning in life and mood

Summarizing conceptual definitions of meaning in life that have been offered in the literature, King and colleagues [17, p. 180] defined meaning in life as follows: "Lives may be experienced as meaningful when they are felt to have a significance beyond the trivial or momentary, to have purpose, or to have a coherence that transcends chaos." Consistent with this definition, most definitions of meaning in life include at least three components, significance (the feeling that one's life and contributions matter to society), purpose (having one's life driven by the pursuit of valued goals), and coherence or comprehensibility (the perception that one's life make sense) [18–20]. These three lower-order components feed into the global experience of meaning in life [21], the focus of the current research. Typically measured with questionnaires, including items like, "I understand my life's meaning," and "My life has a clear sense of purpose" [22], self-reported meaning in life is associated, prospectively, with positive life outcomes across a variety of domains, including lower suicide risk, greater lifetime earnings, better survival rates after health crises, and reduced mortality [23]. Although meaning in life is relatively stable, it is a subjective feeling that can be affected by manipulations, including exposure to stimuli that make sense [24], reminders of the reality of death [25,26], social exclusion and ostracism [27] and mental simulation [28]. Thus, meaning in life can fluctuate in response to changing circumstances [29]. Among the changing circumstances that affect meaning in life are those that affect mood, as we now consider.

Positive and negative affect refer to mood states with either pleasant or unpleasant hedonic valence and varying in arousal [30,31]. For example, positive affect involves feeling happy, content, or ecstatic; negative affect involves feeling sad, worried, or enraged. Positive affect is a robust correlate of meaning in life [17]. Experience sampling studies show that daily positive affect predicts higher meaning in life [17,32], and daily positive affect is a stronger predictor of meaning in life than goal-related thought and activity [17]. In addition, experimentally induced positive affect increases meaning in life [17]. Negative mood is negatively related to meaning in life in cross-sectional data [17] but daily negative mood is less strongly related to

meaning in life, compared to daily positive mood [32], and induced negative affect does not affect meaning in life [17].

In addition, positive mood can serve a compensatory function, bolstering meaning in life in the absence of other factors that typically promote it, such as religiosity [33], social relationships [34,35], and financial resources [36]. Specifically, people who are high in (or primed with) such resources, tend to report relatively high meaning in life, regardless of mood. In contrast, those low on these resources (or who are not primed with them) may report high levels of meaning in life if they are in a good mood [23]. Essentially, this means that positive affect does not predict meaning in life among those high in (or primed with) putative sources of meaning (because it tends to be high across levels of mood). In contrast, positive affect more strongly predicts meaning in life among those who are low on a putative source of meaning (or in control conditions).

Although clearly related, previous research shows that existential and affective concerns are separable responses to experimental manipulations. Exposure to stimuli that make sense boosts meaning in life but has no effect on mood [24]. In addition, mortality salience manipulations bolster worldviews and self-esteem (existential defenses) even controlling for mood [6].

## Right-wing authoritarianism, meaning, and mood

Individual differences in right-wing authoritarianism relate to meaning in life and affect differently. Right-wing authoritarianism is correlated positively with meaning in life [37,38]. A series of studies showed that this correlation is not fully explained by personality traits, cognitive ability, information processing styles, or demographic, ideological, political, religious, or well-being variables [39].

The positive association between right-wing authoritarianism and meaning in life sets this ideology apart from a closely related construct, social dominance orientation. Modern perspectives often characterize the authoritarian personality as involving two dimensions, right-wing authoritarianism and social dominance orientation [40]. Social dominance orientation reflects preference for group-based hierarchies and endorsing the use of force to maintain them [41]. Right-wing authoritarianism and social dominance orientation each result from unique developmental pathways and predict unique outcomes [5,39]. Most importantly for our purposes, social dominance orientation is unrelated to meaning in life [39], and thus is not a focus of the current research.

Among well-being constructs, the positive association with right-wing authoritarianism appears to be unique to meaning in life. Womick and colleagues (2019) found that a small association between life satisfaction and authoritarianism was explained by meaning in life [39, see also 42]. Moreover, right-wing authoritarianism is not related to positive mood [39,43]. It is correlated, instead, with markers of negative affect including trait anger and hostile attributional style [44]. Interestingly, right-wing authoritarianism moderates the association between distress and meaning in life, such that those high on authoritarianism maintain a higher level of meaning in life in the presence of distress [39], demonstrating that the positive association between authoritarianism and meaning in life maintains in the presence of negative affect.

The positive association of right-wing authoritarianism with meaning in life and the negative association of right-wing authoritarianism with mood, suggest that these outcomes may be affected differently by exposure to authoritarian ideals. If these correlational relationships have causal underpinnings, they directly inform expectations for the current research: Exposure to right-wing authoritarian ideals may enhance meaning in life even as they dampen mood.

Thus, we expected expressions of right-wing authoritarianism to enhance the experience of meaning in life, despite their effect on mood. In addition to past research, historical and contemporary theoretical perspectives also suggest an existential function of exposure to authoritarian values, as we now consider.

## The existential function of authoritarian values

Existential and psychodynamic scholars have long noted that authoritarian values offer a solution to the problem of meaninglessness. Frankl [45] proposed that the erosion of typical sources of meaning, such as religion, led people to perceive their lives as relatively meaningless. Frankl argued that this existential vacuum might render individuals more susceptible to authoritarianism. Similarly, Fromm [46] suggested that, in embracing individual freedom (vs. societal controls), humans run the risk of adopting allegiance to authoritarian leaders to ward off uncertainty. In the absence of worldviews that offer commensurate existential security, the experience of freedom from societal controls can move people to seek solace in authoritarian ideals.

Contemporary theoretical perspectives also imply that ideological frameworks like right-wing authoritarianism might contribute to meaning in life. First, Significance Quest Theory [47] argues that perceptions of insignificance place individuals at risk for the adoption of extreme beliefs. The theory suggests that full commitment to extreme belief systems and goals enhances one's sense of significance. Interestingly, the correlation between right-wing authoritarianism and global meaning in life is explained by feelings of significance, rather than purpose or coherence [39].

Second, Terror Management Theory (TMT) [6] argues that people endorse worldviews like right-wing authoritarianism to gain a sense of symbolic immortality. Specifically, endorsement of cultural worldviews and enacting behavior consistent with these facilitates the sense that members of society value one's life and one's contributions. TMT research suggests that worldviews may also contribute to a sense of meaning through existential significance. After a reminder of their own mortality, people high (vs. low) on right-wing authoritarianism showed more negative evaluations of targets with dissimilar attitudes [3,6] suggesting the existential relevance of authoritarian ideals.

Finally, the Theory of Conservatism as Motivated Social Cognition [48] argues that needs to reduce threat and ambiguity underlie the adoption of conservative views such as right-wing authoritarianism. In support of this theory, research shows that people high on conservative beliefs tend to be higher on constructs like need for cognitive closure [49], personal need for structure [50], intolerance of ambiguity [51], and lower in integrative complexity [52]. If right-wing authoritarianism compensates for cognitive deficits, then exposure to authoritarian values may enhance meaning because they convey a cognitively certain view of an unambiguously ordered and structured world.

These theoretical models all imply that right-wing views might serve a causal existential function but this link has not been tested empirically. Here, we provide novel experimental tests of this fundamental theoretical assumption.

## Overview of studies

To the best of our knowledge, no previous research has examined whether exposure to right-wing authoritarian consistent information affects mood or meaning in life. The lack of precedent for these studies led us to adopt a strategy of conceptual and direct replications and focus on main effects. Building directly on previous correlational research demonstrating positive links between right-wing authoritarianism and meaning in life, but not mood [39], five studies

tested the prediction that exposure to right-wing authoritarian values would enhance meaning in life. In all but Study 1, we also tested the prediction that exposure to right-wing authoritarian messages would worsen mood. In Study 1, we administered measures of meaning in life and right-wing authoritarianism in counterbalanced order to test the prediction that meaning in life would be higher when rated after (vs. before) right-wing authoritarianism.

Studies 2 through 5 employed authoritarian passages, egalitarian passages, and a control passage. We predicted that reading an authoritarian message, whether from a real-world authoritarian leader (Study 2), or in passages we authored (Studies 2–5) would lead to worse mood but higher meaning in life compared to a control (or egalitarian) passage. Studies 3 and 4 were preregistered.

An egalitarian values condition was included in Studies 2–5 to determine whether authoritarian values uniquely affect meaning in life and mood relative to a control, or if exposure to a statement of any framework of values would demonstrate a similar effect. The theoretical perspectives reviewed above implicate an existential function of right-wing authoritarianism, but no such function has been hypothesized to exist for egalitarian values. Instead, left-wing ideology is positively related to dispositional positive affect (but not lower negative affect) [53]. Political liberalism (typically operationalized as the opposite of conservatism) is negatively related to meaning in life [39,42]. Thus, we expected exposure to statements of egalitarian values might enhance mood but, in contrast to authoritarian values, would not boost meaning in life, relative to a control.

In all studies, we tested predictions for each type of outcome controlling for the other to probe the effects of condition on affect only (controlling for meaning in life) and meaning in life only (controlling for affect). Given contemporary issues with replicating findings [54], we sought consistent evidence for the effect of exposure to authoritarian values on mood and meaning in life. We also tested whether condition effects on mood and meaning in life would predict evaluations of the passages. Finally, we report exploratory analyses probing the potential interaction of mood and condition in the Results for meaning in life. These analyses also address whether negative affect in response to the authoritarian messages might promote meaning in life. The chronological order of the studies differs from their presentation here to enhance the coherence of the report.

## Study 1

We first sought to demonstrate the viability of the hypothesis that exposure to right-wing authoritarian ideas would enhance meaning in life in a simple study. Participants rated right-wing authoritarianism and meaning in life, with about half rating right-wing authoritarianism first and the other half rating meaning in life first. We predicted that meaning in life would be higher for those who completed the right-wing authoritarianism scale first (compared to those who completed the meaning in life measure first), and that this difference would obtain controlling for levels of authoritarianism espoused on the right-wing authoritarianism scale. We assumed that the effect size for this subtle manipulation would be small and sought to recruit at least 500 participants per condition.

### Method

#### Participants

Participants were 1,053 Amazon Mechanical Turk (MTurk) workers. The sample was 63.8% women, 76.3% white/European American, 9.7% Black/African American, 6.6% Asian American, 5.5% Hispanic/Latinx, and 1.9% indicated "other." Age, *M (SD)* = 37.62 years (12.86) ranged from 18 to 74. Modal education was a bachelor's degree. Median income was $50,001-

$75,000. The study took about 2 minutes, and participants were paid $0.05. Data collection occurred from March 8, 2019–March 11, 2019. All studies were approved by the Institutional Review Board at the corresponding author's institution and were conducted in an ethical manner.

## Measures and procedure

In this and all studies, sample sizes were determined before conducting data analyses, and we report all measures, manipulations, and exclusions. All scales ranged from 1 (*strongly disagree/ not at all*) to 7 (*strongly agree/very much*). Participants completed several measures of meaning in life. For global perceptions of meaning, they completed the 5-item Presence of Meaning in Life Questionnaire (MLQP; e.g., "I understand my life's meaning,") [22], *M (SD)* = 4.98 (1.36), α = .92, the most widely used and validated measure of meaning in life [55]. The other 5-item subscale of the MLQ measures the search for meaning in life, which was not relevant to the current research questions. High endorsements of search for meaning in life do not reflect the evaluation of life as meaningful and low scores do not necessarily reflect meaninglessness (search for and presence of meaning in life have been found to be positively and negatively related, see [56]).

In addition, participants completed the Tripartite Meaning Scale [21], which measures purpose (sample item: "I have a good sense of what I'm trying to accomplish in life"), *M (SD)* = 5.20 (1.24), α = .83, coherence (e.g., "I can make sense of the things that happen in my life"), *M (SD)* = 5.10 (1.30), α = .84, and existential significance ("Whether my life ever existed matters even in the grand scheme of the universe"), *M (SD)* = 5.12 (1.47), α = .77, as well as global meaning in life, (e.g., "My entire existence is full of meaning"), *M (SD)* = 5.54 (1.34), α = .90.

Participants completed the 14-item Right-wing Authoritarianism: Aggression, Conventionalism, and Submission scale [57,58] (example item: "The only way our country can get through the crisis ahead is to get back to our traditional values, put some tough leaders in power, and silence the troublemakers spreading bad ideas"), *M (SD)* = 3.29 (1.18), α = .90. This measure shows high convergent validity with other measures of authoritarianism [57].

Approximately, half (*n* = 520) of the participants completed the right-wing authoritarianism scale before the meaning measures, and the rest completed the right-wing authoritarianism scale after meaning measures, (*n* = 552).[3] A sensitivity analysis (using G*Power) [59] with alpha set at.05 indicated that Study 1 provided 80% power to detect an effect size of *d* = 0.17. In all studies, in line with our pre-registrations, we did not exclude participants from analyses. Data for all studies can be found online: https://osf.io/me34h/?view_only== ed450f2a71e345b59dd6418f5c2821ca.

## Results

### Preliminary analyses

Collapsing across conditions, all of the meaning in life measures were positively related, *r* (1024–1053)'s ranged from.64 for significance and purpose to.81 for the MLQ-P and general meaning measure from the Tripartite Meaning Scale, all *p*'s < .001. Right-wing authoritarianism was positively correlated with MLQP, *r*(1022) = .17, *p* < .001, general meaning *r*(1038) = .13, *p* < .001, and purpose, *r*(1038) = .07, *p* = .024, significance *r*(1038) = .21, *p* < .001, and coherence *r*(1038) = .13, p < .001. Replicating past research (Womick et al., 2019), the second order partial correlation between significance and right-wing authoritarianism (controlling for purpose and coherence) remained significant, *r*(1034) = .21, *p* < .001. Controlling for significance, wiped out the association between purpose [partial *r*(1035) = -.08] and coherence [partial *r*(1035) = -.07] with right-wing authoritarianism.

**Table 1. Effect of order on meaning in life and right-wing authoritarianism, Study 1.**

| Dependent Measures | Measure completed first | | t | d |
|---|---|---|---|---|
| | Meaning in Life | RWA | | |
| RWA(overall) | 3.26 (1.17) | 3.32 (1.19) | 0.76 | 0.05 |
| Submission | 3.63 (1.30) | 3.68 (1.26) | 0.69 | 0.04 |
| Conventionalism | 2.42 (1.45) | 2.36 (1.48) | 0.64 | 0.04 |
| Aggression | 3.58 (1.68) | 3.68 (1.69) | 0.98 | 0.06 |
| Meaning in Life | | | | |
| MLQ-*P* | 4.90 (1.44) | 5.07 (1.25) | 2.06* | 0.13 |
| Global Meaning in Life | 5.48 (1.38) | 5.61 (1.29) | 1.48 | 0.10 |
| Purpose | 5.11 (1.31) | 5.30 (1.16) | 2.58** | 0.15 |
| Significance | 5.03 (1.52) | 5.21 (1.40) | 1.93 | 0.12 |
| Coherence | 5.00 (1.34) | 5.34 (1.31) | 2.43* | 0.26 |

Note.

*$p < .05$;

**$p \leq .01$.

RWA = Right-wing authoritarianism. MLQ-P = presence of meaning subscale of the Meaning in Life Questionnaire. For *t*-tests, *df* ranged from 1051 to 1020 and are adjusted for violation of equal variances across groups for MLQ-P, coherence, and purpose.

## Primary analyses

As shown in Table 1, results of *t*-tests showed that although some of the differences were not significant, the pattern was consistent: Meaning in life was higher when the scales were completed *after* a measure of right-wing authoritarianism. To summarize these results, we standardized all meaning measures and aggregated them, α = .93. Those who completed the right-wing authoritarianism measure first, *M (SD)* = 0.07 (0.85) reported significantly higher meaning in life than those who completed the meaning in life measures first, *M (SD)* = -0.06 (0.92), $t(1051) = 2.43$, $p = .015$, $d = 0.15$. A repeated measures GLM, treating the three facets of meaning as a within participant factor, showed no significant interaction for facet of meaning X order, $F(2,1050) = 0.05$, $p = .95$; for the main effect of order, $F(2,1051) = 6.84$, $p = .009$, $d = 0.20$. Thus, exposure to authoritarian values affected the three facets similarly. Order had no effect on right-wing authoritarianism (nor its facets).

Scores on right-wing authoritarianism did not affect these differences. Using aggregated meaning in life as the dependent variable, controlling for right-wing authoritarianism, $F(1,1035) = 24.30$, $p < .001$, $d = 0.30$, the effect of order remained significant, $F(1,1035) = 5.61$, $p = .018$, $d = 0.14$. Regressing the composite on the predictors showed main effects of right-wing authoritarianism (β = .13, $p = .002$), and order (β = .07, $p = .018$), and no interaction (β = .03, $p = .76$).

## Brief discussion

Study 1 provides an initial demonstration that exposure to right-wing authoritarianism enhances meaning in life. The effect of the manipulation was small, perhaps because we did not account for the effects of mood. The manipulation also lacked realism. People are likely most often exposed to authoritarian values via written or spoken messages in the news or on social media. Thus, Study 2 sought to conceptually replicate Study 1 using new manipulations and testing predictions for mood.

## Study 2

Study 2 used communications from authoritarian and egalitarian figures. To select stimulus materials for these manipulations, we first brainstormed historical authoritarian and egalitarian leaders and located transcripts of their speeches. Once we identified speeches, we selected excerpts that most parsimoniously expressed authoritarian or egalitarian values. In 2 pilot studies (combined $N = 1,103$) using an array of excerpts, we asked participants to read speeches and then rate right-wing authoritarianism items "as if" they were the speaker (to ensure that authoritarian content was clear). To select speeches that were relatively equal on other dimensions, participants also rated them for familiarity, ease of reading, and the likely year they were written.

For authoritarian speeches, we piloted excerpts from Adolf Hitler, Kim Jong Un, Vince Lombardi, Benito Mussolini, George Patton, and Josef Stalin. For egalitarian speeches, we piloted excerpts by Dietrich Bonhoeffer (a German anti-Nazi contemporary of Hitler), Albert Einstein, Mikhail Gorbachev, Martin Luther King Jr., Greg Popovich, Eleanor Roosevelt, and Bernie Sanders. Full results of the pilot studies are shown in the Supplement (pp. 13–21). Pilot data led us to select excerpts from Adolf Hitler, Kim Jong Un, and George Patton (authoritarian leaders); and Dietrich Bonhoeffer, Mikhail Gorbachev, and Eleanor Roosevelt (egalitarian leaders). Additionally, we crafted authoritarian and egalitarian passages based on the definition of authoritarianism and the items on the scale from Study 1. These passages were pilot tested in a variety of ways (see Supplement, p. 2). In a previous experiment using these (see Supplement, pp. 4–12) we found that they led to worse mood but higher meaning in life.

People are often exposed to authoritarian values in the news or social media, so an online study using written materials is a reasonable context for testing our predictions. We expected that statements of authoritarian values would lower mood (controlling for effects on meaning in life) and enhance meaning in life (controlling for effects on mood). Study 2 was preregistered.

## Method

### Participants

1904 participants in the United States were recruited on MTurk to participate in an online study for $1.00. The study took roughly 8 minutes on average. The sample was 52.6% women, 74.9% white/European American, 9.5% Black/African American, 6.6% Asian, 6.6% Hispanic/Latinx, 0.7% Native American, and 1.7% selected "other." Ages ranged from 18–81, $M$ $(SD) =$ 36.07 (11.49). Incomes ranged from under $15,000 to over $151,000, and median income was $35,001-$50,000. Data collection occurred from November 3, 2018 –November 7, 2018. Modal education was a Bachelor's degree. For this and all subsequent studies, we did not run a priori power analyses because we were not interested in identifying a minimum $N$ to detect an effect. Rather, we sought to power the studies as much as possible.

### Procedure

Participants were instructed:

"We are interested in the ways people perceive and remember information. You will be seeing a response given by someone in a previous study. This person was asked to complete a brief writing task, and we are going to show you their response. The response may be from a very old study, so the language may seem antiquated.

"While reading, try to suspend judgment and just learn about this person's experience. Try to focus on the ideas conveyed without immediately judging the author. Please pay close attention because later we will be testing your memory for the information communicated."

In addition to the speech excerpts, we crafted three passages to use in the study. The authoritarian passage was based on the definition of the construct and items from the right-wing authoritarianism scale [1]. The egalitarian passage was written to offer the opposite perspective of that conveyed in the authoritarian essay. The control essay was written to be of approximately the same length as the others but merely described the importance of having a philosophy of life with no information about the content of the writer's perspective. The passages appear below.

## Authoritarian condition

"I believe that it is important to keep a close, tight-knit group that share similar values, that can be trusted to behave morally, to respect the rules of our country, and to preserve the lasting traditions of our culture. Although some may disagree, I am certain that it is ever important to maintain faith in our common sense principles, and stay true to our righteous path.

In order to ensure our security, I think it is important to get in line behind a strong leader that will oppose evil and eradicate the forces eating away at the moral fiber of our country. In following the leaders' vision, if certain individuals have to surrender some freedom in return for a secure nation, so be it."

## Egalitarian condition

"I believe that it is important to immerse oneself in a diverse group of people that come from different backgrounds and espouse a variety values and moral beliefs. In embracing people that are different from ourselves, we can learn from other cultures and use such knowledge to reinvent the traditions of our country. In my opinion, it is important to think independently. We cannot simply trust the conventions of our society without questioning them and improving them.

In order to continue improving our world, I think we should support open-minded leaders that listen to their constituents and act in accordance with our collective interests. In determining the course of our future, we must remember to maintain regard for equality, human dignity and freedom."

## Control condition

"A philosophy of life refers to an individual's worldview beliefs. These beliefs are composed of attitudes and values that guide one's thoughts and play a part in how one decides to act. Sometimes such belief structures are formed through experiences with one's family, in one's religion, or with one's friends, and other times in school or through personal quests. Importantly, a philosophy of life keeps one connected with a greater culture, and helps individuals interpret and make sense of the world.

In order to make the world a better place, everyone should have a code guiding their actions. In deciding what kind of future we want to make for ourselves, it is important to remember what we believe in, and allow those beliefs to guide our decisions."

Participants were assigned randomly to one of the following conditions: Adolf Hitler ($n$ = 210), Kim Jong Un ($n$ = 208), George Patton ($n$ = 212), our authoritarian passage ($n$ = 209), Dietrich Bonhoeffer ($n$ = 214), Mikhail Gorbachev ($n$ = 214), Eleanor Roosevelt ($n$ = 212), our egalitarian passage ($n$ = 213), or the control passage ($n$ = 212). All passages were presented in English. The speeches can be found in the Supplement (pp. 32–33).

### Measures

Rating scales ranged from 1 (*strongly disagree/not at all*) to 7 (*strongly agree/very much*). After the manipulation, participants completed measures of meaning in life, mood, and message evaluations (in that order), followed by demographics. Participants completed the MLQP from Study 1, *M (SD)* = 4.65 (1.47), $\alpha$ = .93. To measure positive affect (PA) participants indicated the extent to which they currently felt cheerful, happy, pleased, and enjoyment/fun, *M (SD)* = 2.87 (1.77), $\alpha$ = .95. For negative affect (NA), they rated how anxious, frustrated, sad, distressed, angry, afraid, worried and nervous they currently felt, *M (SD)* = 2.13 (1.36), $\alpha$ = .95.

Next, participants rated 4 items evaluating the passage: "How much did you like the author of the passage that you read?" "How intelligent is the writer of the passage?" "Did you find the writer's position to be fair?" and "I agreed with the writer's beliefs in the essay." These items were aggregated into a composite representing positive evaluations *M (SD)* = 4.25 (1.66), $\alpha$ = .93. All studies included an item measuring moral superiority. In each study, it was rated higher in the authoritarian condition, but was unrelated to meaning in life. Results are presented in the Supplement (pp. 58–60). Finally, participants rated the familiarity of the passage ("How familiar did the passage seem to you?"), *M (SD)* = 2.58 (1.73). Correlations among variables, collapsed across cells, are shown in the Supplement (S6 Table in S1 File, p. 27).

Participants also completed measures of right-wing authoritarianism, religiosity, and belongingness at the end of the study to explore as potential moderators and mediators. Belongingness partially mediated the effects reported below, see the Supplement, p. 25. In this and all studies, we tested the existential vacuum hypothesis [10]—that authoritarian values would be especially existentially appealing among those who are low on dispositional sources of meaning in life. Given the exploratory nature of these analyses, we present them in the Supplement (pp. 23–24, 42, and 48).

A sensitivity analysis (using G*Power) [59] with alpha set at.05 indicated that, given the sample size, this study had 80% power to detect an effect side of $d$ = 0.16. Analyses for all studies describe results for mood controlling for meaning in life. This aspect of the analyses diverges from our preregistered plans, which described analyses for meaning in life controlling for mood but not similar plans for mood. To probe the independence of results for each set of outcomes, we control for meaning in life in mood analyses below. All analyses in all studies without controlling for meaning in life showed similar results for mood, presented in the Supplement (for Study 2, p. 28, and for Studies 3–5, p. 48).

### Results

Analyses for specific effects of speakers are presented in the Supplement (p. 22). To test our predictions, we collapsed across the specific speakers to compare authoritarian and egalitarian speeches, our authoritarian and egalitarian messages, and the control message. Table 2 shows the results of one-way analyses of variance (ANOVAs), demonstrating that the authoritarian

**Table 2. Effects of condition on message evaluations and familiarity, Study 2.**

|   | Control | Egalitarian Speeches | Original Egalitarian | Authoritarian Speeches | Original Authoritarian | Effect of Condition |
|---|---------|---------------------|---------------------|------------------------|------------------------|---------------------|
| $n$ | 212 | 640 | 213 | 630 | 209 | |
| ME | 5.24 (1.03)$_a$ | 4.76 (1.30)$_b$ | 5.38 (1.27)$_a$ | 3.14 (1.60)$_d$ | 3.92 (1.55)$_c$ | $F_{(4,1828)} = 176.66,^* d = 1.25$ |
| Fam | 3.20 (1.79)$_a$ | 2.39 (1.59)$_b$ | 3.15 (1.93)$_a$ | 2.20 (1.59)$_b$ | 3.09 (1.86)$_a$ | $F_{(4,1828)} = 27.18,^* d = 0.51$ |

Note.

$^* p < .001$.

Means in the same row with differing subscripts significantly differed, $p < .007$, Bonferroni adjusted. ME = evaluations composite; Fam = familiarity.

speeches were evaluated most negatively among all passages. The authoritarian message we crafted was more positively evaluated than those speeches but still differed significantly from the egalitarian and control messages. With regard to familiarity, participants found the speeches to be less familiar than our original passages, suggesting that these historical documents were not generally familiar to participants.

To test our central predictions, analyses of covariance (ANCOVAs) tested the effect of condition on mood, controlling for meaning in life, and the effect of condition on meaning in life, controlling for mood. Adjusted means for both sets of outcomes are shown in Fig 1. As can be seen, control and egalitarian messages led to higher PA and lower NA. The authoritarian message led to considerably less pleasant mood but higher meaning in life.

Controlling for meaning in life, $F_{(1, 1829)} = 167.57$, $p < .001$, condition significantly affected PA, $F_{(4, 1829)} = 41.88$, $p < .001$, $d = 0.61$; and NA $F_{(4,1829)} = 44.27$, $p < .001$, $d = 0.62$ (the effect for meaning in life in the model with NA as the outcome was $F_{(4, 1829)} = 72.68$, $p < .001$). The authoritarian speeches and our authoritarian passage differed significantly from the other two conditions (and not from each other) on both indicators of mood. They led to lower PA and higher NA, $p < .001$, Bonferroni corrected.

With regard to meaning in life, controlling for mood, for PA $F_{(1, 1828)} = 308.40$, $p < .001$, and NA $F_{(1, 1828)} = 69.99$, $p < .001$, the authoritarian speeches and our authoritarian passage boosted meaning in life, $F_{(4,1828)} = 6.77$, $p < .001$, $d = 0.29$. The authoritarian speeches did not differ from our authoritarian passage, and both led to significantly higher meaning in life than the control passage, and our egalitarian passage, and marginally differed from egalitarian speeches, $p = .051$. Egalitarian passages did not differ from each other or the control condition, all $p$'s $> .38$.

Because estimated means for meaning in life controlling for mood (vs. raw means) were higher in the authoritarian condition and lower in the egalitarian condition, we probed the relations of mood and meaning in life within conditions. The relationship between mood and meaning in life did not significantly differ across conditions (with the sole exception of PA in the original authoritarian vs. egalitarian condition). Full results are in the Supplement (p. 23).

## Brief discussion

Exposure to speeches of real-world authoritarian leaders led to worse mood and negative evaluations, but higher meaning in life than a control perspective. Authoritarian speeches enhanced meaning in life despite the antiquity of the language used by the speakers, the fact that they were mostly translated from other languages, and were brief excerpts from larger speeches. In addition, Study 2 showed that the effects of an original passage (based on the definition of authoritarianism and a scale measuring the construct) on mood and meaning in life were similar to authoritarian speeches. Thus, we used this passage in subsequent studies to operationalize exposure to authoritarian values.

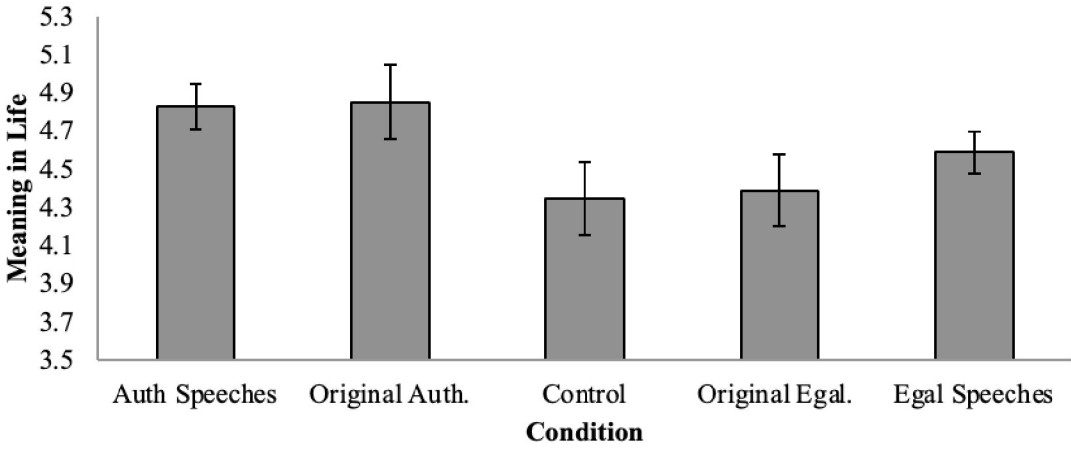

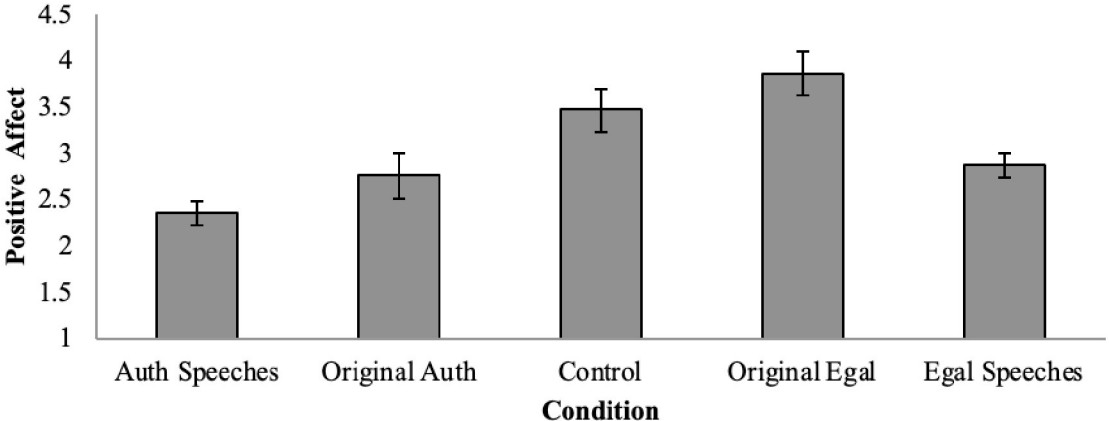

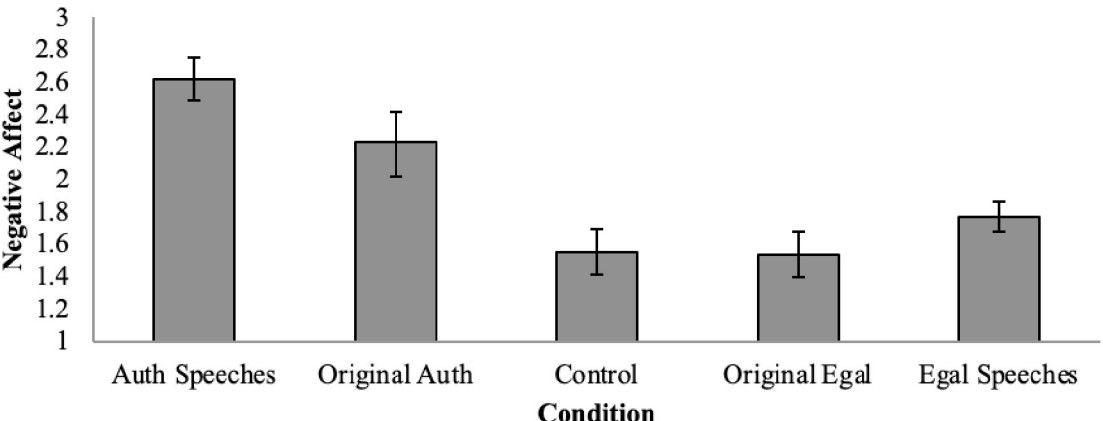

**Fig 1. Effects of condition on meaning in life (top panel), positive affect (middle panel), and negative affect (bottom panel) Study 2.** *Note.* Auth = Authoritarian; Egal = Egalitarian. Error bars are 95% Confidence Intervals.

## Studies 3, 4, and 5

Three studies sought to replicate the pattern identified in Study 2. In all studies, participants read our authoritarian or egalitarian passages, or the control passage from Study 2, then they

rated meaning in life and mood, and evaluated the messages. These data were used to test two *a priori* predictions. First, we predicted that an authoritarian message would lead to lower positive and higher negative mood (controlling for meaning in life) and poorer evaluations of the essays compared to an egalitarian message or a control message. Second, we predicted that controlling for effects on mood, meaning in life would be higher in the authoritarian condition compared to the egalitarian and control conditions. Studies 3 and 4 were preregistered. Our preregistered plan did not include controlling for meaning in life in analyses for mood. Study 5 directly replicated Studies 3 and 4 with people from a less conservative country (Canada). Each study involved independent, separate samples.

If the appeal of authoritarian values is existential, the boost to meaning in life resulting from exposure to them might render authoritarian ideas more appealing. Thus, meaning in life resulting from exposure to authoritarian values could predict subsequent positive evaluations of such messages. We aggregated data from Studies 3–5 to test this possibility probing the role of mood and meaning in life post-manipulation on message evaluations. Finally, we used this aggregated dataset to test for potential interactions among mood and condition predicting meaning in life.

## Common method

### Participants

For all studies, because we assumed that the effect of interest was likely to be relatively small, our goal was to collect at least 500 participants per condition. People who took part in any given study were blocked from other studies. The studies took roughly 5 minutes on average.

**Study 3.** 1639 participants in the United States were recruited on MTurk and participated in an online study for $0.15. Data were collected from November 17, 2017 –November 21, 2017. The sample was 65% women, 75.2% white/European American, 10.2% Black/African American, 5.5% Asian, 6.5% Latino(a), 0.7% Native American, and 2% indicated "other." Ages ranged from 18–76, *M (SD)* = 36.95 years (11.86). Modal education was a Bachelor's, and 88.8% of participants had completed some college or more. Median income was $35,001-$50,000, and incomes ranged from under $15,000 to over $150,000.

**Study 4.** 1607 participants in the United States completed an online study on MTurk for $0.15. Data collection occurred from December 4, 2017 –December 12, 2017. The sample was 64% women, 74.8% white/European American, 9% Black/African American, 6.5% Asian, 6.5% Latino(a), 0.6% Native American and 2.6% indicated "other." Age, *M (SD)* = 35.64 years (11.9), ranged from 10–83. Median income was $35,001-$50,000, and incomes ranged from under $15,000 to over $150,000. Modal education was "some college" and 89.3% of participants had completed some college or more.

**Study 5.** 148 Canadian participants completed an online study via MTurk for $0.15. Data were collected from November 3, 2017 –December 17, 2017. The sample was 52% women. Represented race/ethnicities included, 69.2% white, 4.8% Black, 14.4% Asian, 2.7% Latino(a), and 8.9% "other." Age *M (SD)* = 31.62 years (10.44), ranged from 18–67. Median income was $50,001-$75,000, with incomes ranging from under $15,000 to over $150,000. Modal education was a Bachelor's, and 84% of participants had completed some college or more. We had hoped to collect a much larger sample but after several months of data collection, new Canadians did not enroll in the study. As expected, this sample was significantly less conservative than the American samples (See Supplement, p. 41).

## Procedure and measures

Procedures were the same as in Study 2 (dropping the speeches conditions and the statement from the instructions, "The response may be from a very old study, so the language may seem antiquated"). Participants were assigned randomly to one of the three conditions (authoritarian, egalitarian, or control) and then completed the dependent measures. All participants completed the MLQP (α's ≥ .91), PA (α's >.94) and NA (α's>.89) measures, and message evaluations (α's >.88) from Study 2. In Study 5 only, at the end of the survey, participants completed the Right-wing Authoritarianism measure from Study 1, *M (SD)* = 2.58 (1.00), α = .89. Participants in Study 5 also completed a brief measure of intrinsic religiosity (2 items from the Revised Intrinsic Religiosity Orientation Scale, [60]; validated in [61]) after the measurement of the dependent variable and covariates. This variable was tested as moderator of condition effects on meaning in life. Generally, results were not significant. Full description can be found in the Supplement (pp. 41–47).

All participants completed demographic questions (including religious affiliation) and a single item rating of political orientation, ranging from 1 (*very liberal*) to 7 (*very conservative*). (Note, conservatism interacted with condition to predict meaning in life. These results emerged because conservatives were at a ceiling on meaning in life, and liberals lack this dispositional ideological source of life's meaningfulness (Newman et al., 2019). Results are presented in the Supplement, p. 48–50).

Correlations among variables are shown in the Supplement (S9 Table in S1 File, p. 36, S10, p. 39, and S11, p. 43 for Studies 3, 4 and 5, respectively). A sensitivity power analysis for each sample with alpha set at.05 [59], indicated that we had 80% power to detect effect sizes of $d = 0.14$ (Study 3), $d = 0.14$ (Study 4), and $d = 0.50$ (Study 5).

## Results

Table 3 shows the results for one-way ANOVAs on message evaluations. As in Study 2, the authoritarian passage was evaluated more poorly than the other two passages.

With regard to the main dependent measures, we again tested our prediction using ANCOVAs. Fig 2 shows the means for each type of outcome controlling for the other across all three studies. For mood, the pattern of results was similar across studies. All pairwise comparisons are Bonferroni corrected. For PA, condition effects were significant, for Study 3, $F(2, 1563) = 51.75, p < .001, d = 0.51$; for Study 4, $F(2, 1716) = 103.63, p < .001, d = 0.70$; and for Study 5, $F(2, 144) = 11.10, p < .001, d = 0.79$. In Studies 3 and 4 the authoritarian condition led to

**Table 3. Effects of condition on message evaluations, studies 3, 4, and 5.**

| Study | | Control | Egalitarian | Authoritarian | Effect of Condition |
|---|---|---|---|---|---|
| 3 | *n* | 519 | 527 | 527 | |
| | ME | 5.21 (1.16)$_a$ | 5.32 (1.34)$_a$ | 3.98 (1.74)$_b$ | $F(2,1562) = 150.74,$** $d = 0.87$ |
| 4 | *n* | 505 | 506 | 501 | |
| | ME | 5.26 (1.07)$_a$ | 5.32 (1.27)$_a$ | 3.95 (1.47)$_b$ | $F(2,1504) = 182.68,$** $d = 1.00$ |
| 5 | *n* | 46 | 52 | 50 | |
| | ME | 4.84 (1.15)$_a$ | 5.43 (0.76)$_b$ | 4.03 (1.43)$_c$ | $F(2,145) = 19.29,$** $d = 1.03$ |

*Note*. ME = Message Evaluation.

[†]$p = .01$;

[*]$p = .004$;

[**]$p < .001$.

Means in the same row with differing subscripts are significantly different, $p < .05$, Bonferroni corrected.

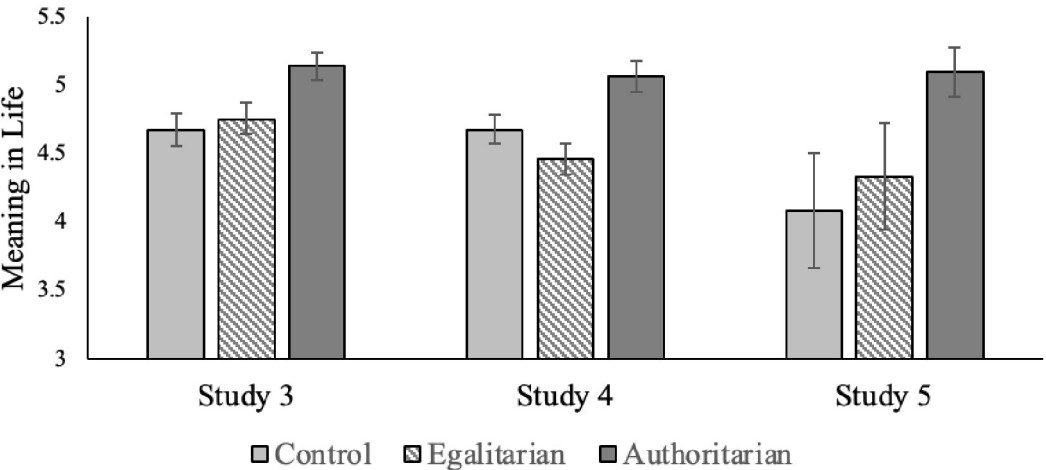

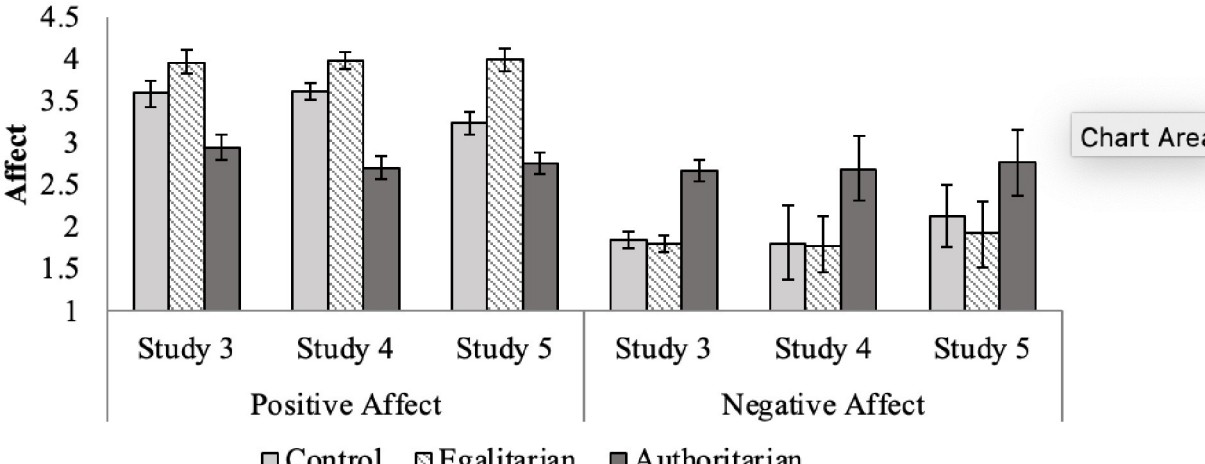

**Fig 2. Condition effects on meaning in life (controlling for mood; top panel) and mood (controlling for meaning in life; bottom panel), Studies 3–5.** *Note.* Messages were descriptions of egalitarian or authoritarian worldviews or an essay describing the importance of having a philosophy of life (control). Participants in Studies 3 and 4 were Americans. Study 5 was a Canadian sample. PA = positive affect; NA = negative affect. Error bars are bootstrapped 95% confidence intervals with 1000 re-samplings. In the models with PA as the outcome, we controlled for meaning in life. In these models, the effects for meaning in life were, Study 3, $F(1, 1563) = 146.13$, $p < .001$; Study 4, $F(1, 1509) = 127.64$, $p < .001$; and, Study 5 $F(1, 144) = 19.45$, $p < .001$. For the models considering NA as the dependent variable, we controlled for meaning in life. The effects for meaning as a covariate were as follows: Study 3, $F(1, 1563) = 50.81$, $p < .001$; Study 4, $F(1, 1508) = 56.74$, $p < .001$; and, Study 5 $F(1, 144) = 1.22$, $p = .27$. In the models considering meaning in life as the outcome, the effects for PA were in, Study 3, $F(1, 1562) = 144.27$, $p < .001$; Study 4, $F(1, 1507) = 126.52$, $p < .001$; and, Study 5 $F(1, 143) = 19.26$, $p < .001$; and NA, Study 3, $F(1, 1562) = 49.10$, $p < .001$; Study 4, $F(1, 1507) = 55.81$, $p < .001$; and, Study 5 $F(1, 143) = 1.16$, $p = .18$.

significantly lower positive mood than the other two conditions (which did not differ from each other), $p$'s $\leq .001$. For Study 5, the egalitarian condition led to higher PA than the other conditions, $p < .001$, which did not differ from each other (likely due to the small sample size, as the pattern was the same as in the other studies).

For NA, controlling for meaning in life, condition effects were significant, for Study 3, $F(2,1563) = 69.79$ $p < .001$, $d = 0.60$; for Study 4, $F(2,1715) = 90.67$, $p < .001$, $d = 0.70$; and for Study 5, $F(2,144) = 5.12$ $p = .007$, $d = 0.53$. In Studies 3 and 4, the authoritarian condition led to higher NA than the other two conditions. In Study 4, the egalitarian condition led to higher NA than the control condition. In Study 5, the authoritarian condition was higher in NA than

Table 4. Effects of condition on meaning in life, Studies 3, 4, and 5.

| | | Control | Egalitarian | Authoritarian | Effect of Condition & Effect of Condition controlling for mood |
|---|---|---|---|---|---|
| **Study 3** | *n* | 519 | 527 | 527 | |
| Raw *M (SD)* | | 4.72 (1.37) | 4.90 (1.35) | 4.93 (1.32) | $F(2, 1570) = 3.78^*, d = 0.14$ |
| *M* adjusted for mood | | $4.67_a$ | $4.75_a$ | $5.14_b$ | $F(2,1562) = 18.67^{**}, d = 0.29$ |
| **Study 4** | *n* | 505 | 506 | 501 | |
| Raw *M (SD)* | | 4.78 (1.24) | 4.62 (1.32) | 4.85 (1.36) | $F(2, 1513) = 4.13^*, d = 0.14$ |
| *M* adjusted for mood | | $4.69_a$ | $4.46_b$ | $5.11_c$ | $F(2,1507) = 30.95^{**}, d = 0.41$ |
| **Study 5** | *n* | 46 | 52 | 50 | |
| Raw *M (SD)* | | 4.00 (1.42) | 4.58 (1.43) | 4.91 (1.33) | $F(2, 145) = 5.21^{**}, d = 0.54$ |
| *M* adjusted for mood | | $4.08_a$ | $4.33_a$ | $5.10_b$ | $F(2,143) = 7.45^{**}, d = 0.63$ |

Note.

$^*p < .024$;

$^{**}p < .008$.

Means in the same row with differing subscripts are significantly different, $p < .05$, Bonferroni corrected.

the egalitarian condition but did not differ from control (again, possibly due to the small sample size).

Table 4 shows results for meaning in life across all studies. In accord with our preregistered prediction, controlling for mood, exposure to a statement of authoritarian values led to higher meaning in life than egalitarian or control perspectives. In Studies 3 and 5 the control and egalitarian conditions did not differ from each other. In Study 4, the egalitarian condition was significantly lower than the control condition.

Although we had preregistered our intent to control for mood, we conducted a sensitivity analysis, testing for condition effects in the absence of controlling for mood. As Table 4 shows condition effects were significant, without accounting for mood. Similarly, controlling for all covariates, condition effects on meaning in life remained significant: Study 3 $F(2,1558) = 11.14$, $p < .001$, $d = 0.24$; Study 4 $F(2, 1497) = 22.92$, $p < .001$, $d = 0.35$; Study 5 $F(2, 141) = 5.37$, $p = .006$, $d = 0.55$. Corresponding density plots can be found in the Supplement (S6 Fig in S1 File, p. 37, S7, p. 40, and S8, p. 44, for Studies 3, 4 and 5, respectively).

Comparing the raw means to those adjusted for mood shows that in every case, controlling for mood lowered the means for meaning in life in the egalitarian condition and increased meaning in life in the authoritarian condition. To test whether the slopes for the mood variables varied across conditions, within each dataset, we computed hierarchical regression models. We mean-centered PA, NA, and computed dummy variables representing the authoritarian (1 = authoritarian, 0 = other), and egalitarian conditions (1 = egalitarian, 0 = other). Main effects were entered on the first step, and the two-way interactions were entered on the second step. Across the three studies, the mood X condition step contributed significantly to meaning in life only in Study 3, $\Delta R^2 = .023$, $p < .001$. For Study 4, $\Delta R^2 = .005$, $p = .15$; for Study 5, $\Delta R^2 = .022$, $p = .58$. Within Study 3, the slopes for PA did not differ across conditions, $z$'s $< 0.99$. For NA, the slopes across condition were, $b$ (*SE*) = -0.17 (0.04) for authoritarian; $b$ (*SE*) = -0.17 (0.05), for egalitarian; and $b$ (*SE*) = -0.28 (0.04) for the control group. Clearly, the slopes for authoritarian and egalitarian were identical. The association between NA and meaning in life was significantly stronger in the control condition than the authoritarian condition, $z = 2.21$, $p = .027$. The difference between the control and egalitarian condition was marginal, $z = 1.88$, $p = .06$. In sum, the slopes of mood predicting meaning in life within condition did not vary between the authoritarian and egalitarian conditions. We probe this issue further below.

## Do mood or meaning in life, post-manipulation, predict later message evaluations?

Next, we tested whether the influence of condition on mood and meaning in life might, in turn, influence message evaluations (completed after the meaning in life and mood ratings) as a function of message content. We merged the data from Studies 3–5 ($N = 3401$) for these regression analyses. Re-analyses testing our central predictions with the pooled data showed the same pattern of results (see the Supplement, p. 61).

Collapsed across conditions, PA and meaning in life were positively related to message evaluations, $r's$ (3220) = .56 and .11, respectively $p's < .001$. NA was negatively related to message evaluations, $r = -.11$, $p < .001$. To examine whether mood or meaning in life related to message evaluations differently across conditions, we computed hierarchical regression equations, entering mean-centered PA, NA, or meaning in life, dummy variables representing the authoritarian (1 = authoritarian, 0 = other), and egalitarian conditions (1 = egalitarian, 0 = other) on the first step, and the continuous variables X condition interactions on subsequent steps.

For mood, message evaluations were regressed on the main effects of PA, NA, and the condition dummies ($\Delta R^2 = .47$, $p < .001$) on the first step, all possible two-ways on the second step ($\Delta R^2 = .01$, $p < .001$), and the terms representing the three-way interaction on the final step, ($\Delta R^2 = .003$, $p < .001$). Although the three-way step was significant, decomposing this interaction within each condition revealed, essentially, the straightforward main effects of differently valenced mood—with the main effects of PA being positive in the egalitarian ($\beta = .54$) control ($\beta = .44$) and authoritarian ($\beta = .53$) conditions, all $p's < .001$. Similarly, the main effects of NA were negative across conditions: $\beta = -.33$ for egalitarian; $\beta = -.21$ for control, and $\beta = -.27$ for the authoritarian conditions, all $p's < .001$. Thus, positive affect predicted more positive evaluations and negative affect predicted more negative evaluations, regardless of message content.

Results for mood are straightforward. Does the same hold true for meaning in life? To answer this question, we regressed evaluations hierarchically on mean-centered meaning in life and the condition dummy variables on the first step and the meaning in life X condition interactions on the second step. For the first step, $\Delta R^2 = .19$, $p < .001$, the main effects for meaning in life, $\beta = .13$, $p < .001$, and authoritarian condition, $\beta = -.41$, $p < .001$, contributed significantly; the egalitarian condition did not, $\beta = .03$, $p = .09$. These main effects were qualified by significant meaning in life X authoritarian condition ($\beta = .06$, $p = .011$) and meaning in life X egalitarian interactions ($\beta = -.06$, $p = .012$), entered on the second step, $\Delta R^2 = .01$, $p < .001$. Generated regression lines for those +/- 1 $SD$ from the mean on meaning in life in each condition are shown in Fig 3. As can be seen, meaning in life was positively related to evaluations, especially in the authoritarian condition (with an apparent ceiling effect on evaluations in the other conditions). The link between meaning in life and evaluations was stronger in the authoritarian condition ($\beta = .22$, $p < .001$) vs. the egalitarian condition ($\beta = .04$, $p = .21$), $z = 4.25$, $p < .001$; but it did not differ from control ($\beta = .17$, $p < .001$), $z = 1.25$, $p = .21$. These results lend some support the idea that meaning in life following exposure to authoritarian values predicts more positive evaluations of the message. In addition to these results for meaning in life, PA significantly interacted with condition to predict evaluations. Results are in the Supplement (pp. 51–55).

## Brief discussion

As predicted, exposure to a statement of authoritarian values consistently led to lower PA and higher NA and higher meaning in life than egalitarian and control statements, despite engendering negative evaluations. The mood effects of condition led to more positive evaluations for messages that boosted PA and more negative evaluations for messages that enhanced NA,

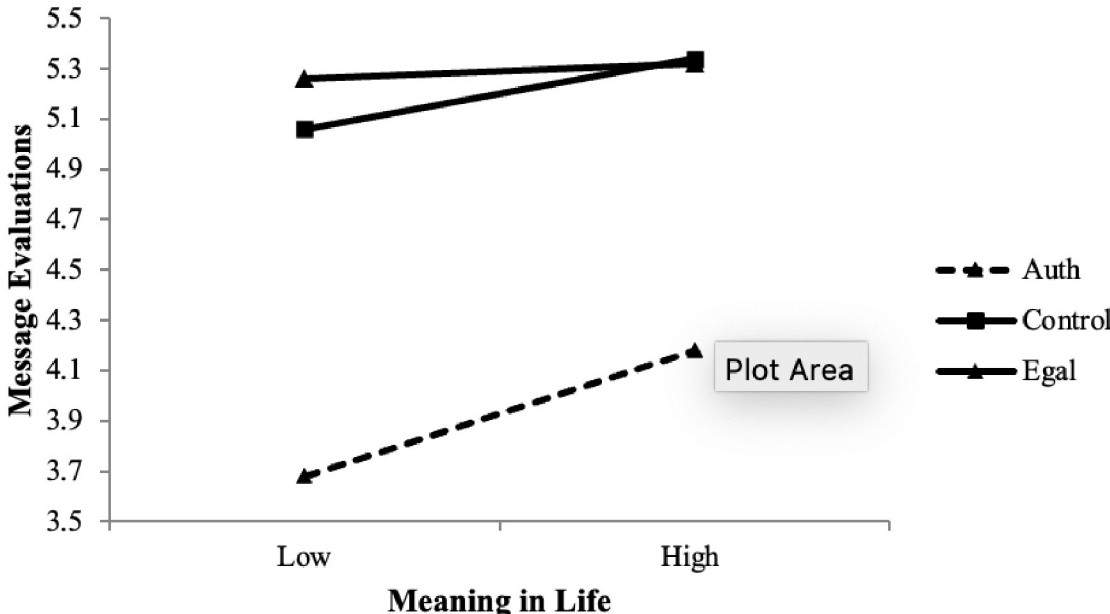

**Fig 3. Interaction of post-manipulation meaning in life X authoritarian dummy predicting message evaluations, pooled data, Studies 3–5. Note**. The slope for the authoritarian condition is significantly stronger in the authoritarian condition than the egalitarian condition but does not differ from the control condition.

regardless of the content of the message. However, meaning in life after exposure to authoritarian values predicted more positive subsequent evaluations of the authoritarian passage. This effect of meaning in life was unique to the authoritarian condition. The pattern of results suggests that this unlikeable worldview may become (even slightly) more likeable as a function of its effects on meaning in life.

### Did mood moderate condition effects on meaning in life?

As noted previously, in every study, controlling for mood led to lower adjusted (vs. raw) meaning in life in the egalitarian condition and higher adjusted (vs. raw) meaning in life in the authoritarian condition. Although analyses within each dataset did not suggest that mood played differing roles across these conditions, we probed this issue further, using the pooled dataset. We hierarchically regressed meaning in life on dummy codes for authoritarian and egalitarian conditions, along with mean-centered PA and NA, and all 2-way, and the 3-way interactions. As Table 5 shows, the three-way interaction step contributed significantly to the equation, and both three-way interaction terms were significant.

To decompose these interactions, we examined each condition separately. In the control condition, PA ($\beta$ = .41 $p$ < .001) and NA ($\beta$ = -.30, $p$ < .001) both contributed significant main effects (for main effect step, $\Delta R^2$ = .21, $p$ < .001); the PA X NA interaction was also significant, $\Delta R^2$ = .01, $\beta$ = .07, $p$ = .012. For the authoritarianism condition, PA ($\beta$ = .29 $p$ < .001) and NA ($\beta$ = -.18, $p$ < .001) both contributed significant main effects (for main effect step, $\Delta R^2$ = .10, $p$ < .001); the PA X NA interaction was also significant, $\Delta R^2$ = .02, $\beta$ = -.14, $p$ < .001. For the egalitarianism condition, PA ($\beta$ = .21 $p$ < .001) and NA ($\beta$ = -.12, $p$ < .001) both contributed significant main effects (for the main effects step, $\Delta R^2$ = .07, $p$ < .001); the PA X NA interaction was not significant, $\beta$ = -.04, $p$ = .31.

With regard to the main effects of mood within each condition, PA was more strongly associated with meaning in life in the control condition compared to the authoritarian condition,

**Table 5. Probing moderation of condition effects on meaning in life by mood.**

| Variables Entered on Step | $\Delta R^2$ for Step | β |
|---|---|---|
| **Step 1, Main Effects** | .12** | |
| Authoritarian Dummy | | .17** |
| Egalitarian Dummy | | .01 |
| Positive Affect (PA) | | .42** |
| Negative Affect (NA) | | -.34** |
| **Step 2, 2-way interactions** | .012** | |
| Authoritarian X PA | | -.07* |
| Authoritarian X NA | | .12** |
| Egalitarian X PA | | -.11** |
| Egalitarian X NA | | .10** |
| PA X NA | | .08* |
| **Step 3, 3-way interactions** | .005** | |
| Authoritarian X PA X NA | | -.14** |
| Egalitarian X PA X NA | | -.06* |

**Note**. Multiple $R^2$ for the equation = .13, $F(11, 3215) = 44.79$, $p < .001$.

**$p < .001$;

*$p \leq .02$.

Tolerance levels for predictors entered on Steps 2 and 3 ranged from .95 for the PA X NA interaction to .37 the 3-way involving the authoritarian dummy (likely because of the effect of the authoritarian condition on mood).

$z = 2.90$ and the egalitarian condition, $z = 4.12$, both $p$'s $< .001$. The authoritarian and egalitarian conditions did not differ, $z = 0.12$. NA was more strongly related to meaning in life in the control condition than the egalitarian condition, $z = 4.12$, $p < .001$; but did not differ from the authoritarian condition, $z = 1.50$. Generated regression lines predicting meaning in life for those +/- 1 *SD* from the mean on PA within conditions are shown in Fig 4.

Fig 4 shows that meaning in life was higher in the authoritarian condition across levels of PA. Additionally, particularly in comparison to the control condition, meaning in life was relatively high at low levels of PA in the authoritarian condition. This pattern conforms to that found for primes of putative sources of meaning in life described earlier. Similar to the results observed for right-wing authoritarian messages here, when participants are exposed to primes of social relationships [34], they report relatively high levels of meaning in life across levels of PA and in a control condition, PA more strongly predicts meaning in life.

With regard to the PA X NA interactions in the authoritarian and control conditions, the generated slopes for participants who were +/- 1 *SD* from the mean on PA and NA are shown in Fig 5. As can be seen, in comparison to the control condition, poor mood took less of a toll on meaning in life in the authoritarian condition. In the authoritarian condition, meaning in life ratings "bottom out" at a higher value (for those low in PA and high in NA) compared to the controls.

The results in Fig 5 are relevant to an intuitively appealing explanation for the effect of the authoritarian condition on meaning in life. Specifically, feelings of reactance, revulsion, enemyship, or anger [62–64] in response to the authoritarian message might energize people, enhancing meaning in life. However, Fig 5 shows that NA was negatively related to meaning in life in the authoritarian condition, especially at high levels of PA. In the authoritarian condition, meaning in life was highest at high levels of PA and low levels of NA. We further probed this potential reactance explanation with the pooled data set, examining the interaction of

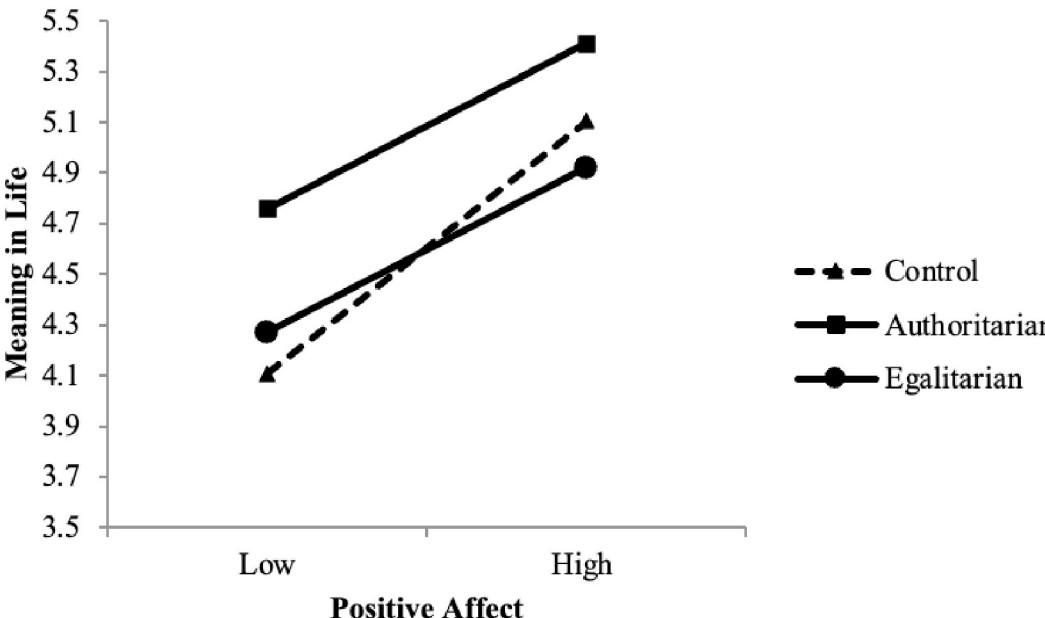

**Fig 4. Predicting meaning in life from PA across conditions. Note**. The slope positive affect predicting meaning in life is weaker than the slope in the control condition but does not differ from the egalitarian condition.

condition with anger and agreement with the message, and found no support for it. Similar to negative affect, in no case did anger or disagreement relate positively to meaning in life (see Supplement, pp. 62–65).

## Mini-meta-analysis

We conducted a mini-meta-analysis comparing the authoritarian and control conditions across all studies [65] for mood (for Studies 2–5) and meaning in life (for all studies). For mood, analyses used means adjusted for meaning in life and raw standard deviations. The average effect size for PA was $d = 0.42(0.039)$, $z = 10.87$, $p < .001$, 95% CI = [0.35, 0.50]. For NA, average $d = -0.59 (0.039)$, $z = 14.98$, $CI = [-0.66, -0.51]$.

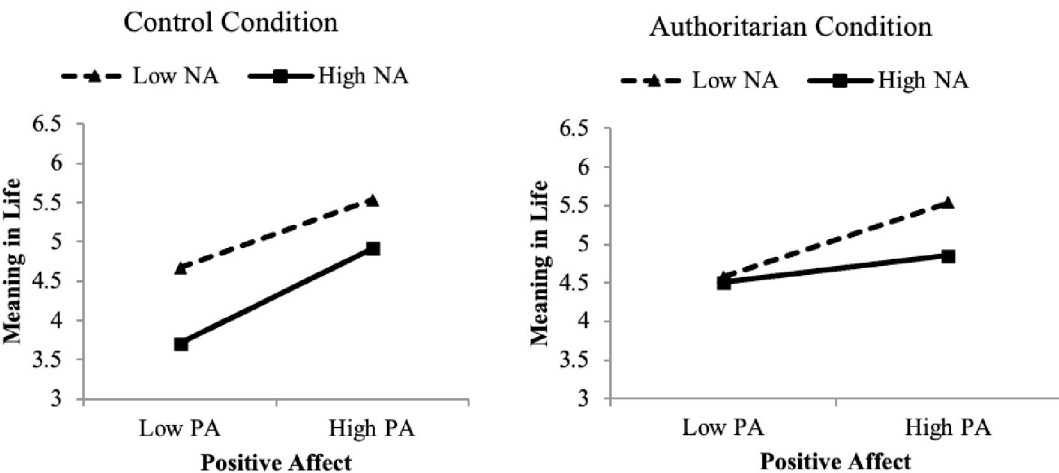

**Fig 5. Positive affect X negative affect predicting meaning in life, in control and authoritarian conditions.**

For meaning in life the mini-meta-analysis used adjusted means (and raw standard deviations) for Studies 2 through 5 and raw values for Study 1. Average $d = 0.28(0.04)$, $z = 7.81$, $p < .001$; 95% CI = [0.21, 0.35]. Without adjusting for mood, results were significant but smaller in magnitude, average $d = 0.13(0.04)$, $z = 3.62$, $p < .001$; 95% CI = [0.06, 0.20].

## General discussion

The proliferation of right-wing authoritarian ideology in contemporary society [9] calls for scientific focus on this important topic. Although there is little question that factors such as societal changes and economic conditions likely play a role in this renewed popularity, the present studies examined how expressions of authoritarian ideology affect important intrapersonal experiences—mood and meaning in life. Across studies, authoritarian messages consistently lowered positive mood and heightened negative mood. At the same time, these messages enhanced the experience of meaning in life. Results for meaning in life were generally consistent whether or not we controlled for mood, but the effect of authoritarian values on meaning in life was stronger when adjusting for affect. In addition, messages that enhanced PA were evaluated more positively; those that enhanced NA were evaluated more negatively, regardless of content. In contrast, meaning in life predicted more positive subsequent evaluations of the message in the authoritarian condition. These findings are consistent with theoretical arguments suggesting that a reason right-wing authoritarianism became widespread in the 20th century was that it served an existential function [45,46].

The present results contribute to a growing literature on the link between ideology and well-being. Numerous correlational studies have shown that conservative ideology is associated with many aspects of well-being [66–69], including life satisfaction [70], and meaning in life [39,42]. The present studies add to this knowledge, showing that exposure to messages conveying one conservative view, right-wing authoritarianism, leads to higher meaning in life while simultaneously lowering mood. Many aspects of these results warrant discussion.

### The experience of meaning in life and acceptance of authoritarian values

Consistent with theoretical perspectives suggesting the meaningfulness of a message should promote belief endorsement [71], Studies 3–5 showed that the existential boost resulting from exposure to authoritarian values predicted more favorable subsequent evaluations of the passage. This finding hints at the possibility that the existentially comforting aspects of authoritarian messages might lead people to see authoritarianism as a viable framework of values. This result may reveal one pathway, among many others [5], by which people come to endorse such views. In the real world, it is possible that individuals who lack a sense of meaning in life may encounter statements of authoritarian values from various sources—viewing posts, videos, or memes on social media sites like 8kun or Twitter, by reading or listening to speeches on the news, in conversations with family or friends, or by attending political rallies. These messages, while putting the person in a foul mood, may lead to a momentary boost in meaning in life that subsequently engenders more favorable evaluations of the ideas.

Although the present results cannot speak to what follows, we speculate that, over time, repeated exposure to authoritarian values, followed by small boosts to meaning in life might lead some people to adopt right-wing authoritarianism as their own framework of beliefs. This process may drive them to seek out and join groups who espouse similar values. Tenets of Significance Quest Theory [72] and TMT [73] suggest that the person would then be motivated to engage in behavior that would result in favorable evaluations by the group, for instance by voting, posting on social media, participating in demonstrations, or possibly more extreme measures, to maintain or build a sense of significance. Future research might probe whether the

increased meaning in life resulting from exposure to authoritarian values might, over time, influence the endorsement of right-wing authoritarianism, itself. Perhaps especially among those experiencing psychological distress [39], lack of personal control [74], or who feel undervalued and unappreciated by society [75,76], the momentary boost to meaning resulting from exposure to authoritarian messages opens the door to adopting this ideology. An authoritarian leader need only capitalize on this opportunity [77]. Future research might consider whether similar processes occur for left-wing authoritarianism [78], or expressions of extreme ideology, in general [79].

Placing the present results in a mood-as-information framework [80], they suggest that authoritarian statements produce a mixed message. Mood consistently provides information rejecting these messages. However, the feeling of meaning [29] presents an opposing view: On the existential level, these messages evoke a more positive response. Specifically, the meaning-as-information approach suggests that life is experienced as meaningful when experiences make sense. Thus, it may be that, when exposure authoritarian values enhances the sense that life is meaningful, it is helping people make sense of their experiences.

Conceptually, the relation of mood and authoritarianism is complex, with negative affect likely playing a role in the appeal of authoritarian values [76]. In attempting to capture the appeal of populism and right-wing views, along with other important factors, scholars have focused on the affective appeal of these for groups. In their rhetoric, populist leaders construct a problem or source for aggrievement and then suggest a solution to that problem [81], potentially engendering positive feelings for the future among people feeling marginalized or put down. Analyses of social media posts concerning political change in the Philippines suggest a mixture of intense positive affect (linked to the promise of change) and strong negative affect (in contempt for those with differing views) [82]. The present studies, focusing on intra-individual processes, may miss the importance of group identity and group-related feelings to the appeal of authoritarianism.

Indeed, although mood and meaning in life are intrapersonal experiences, the spread of authoritarianism is likely deeply embedded in the interpersonal world. Notably, in Studies 2–5, the author of the authoritarian message was not identified as an authority, but simply another participant in a previous study. That such information, coming from an anonymous peer, influenced meaning in life suggests that larger effects might be expected if authoritarian messages were delivered by a family member, loved one, or powerful figure. Authoritarian leaders do not require people to endorse authoritarianism to institute an authoritarian regime. Rather, they require only that people tolerate the idea until it is too late to resist [77].

## Conceptually plausible mediators

Right-wing authoritarianism is a multifaceted construct and our manipulations sought to present a comprehensive representation of this worldview. Results with regard to the prediction of meaning in life by positive mood across conditions suggest that authoritarian messages served a function similar to primes of social relationships or other putative sources of meaning—rendering meaning in life relatively high across levels of positive affect. Exposure to right-wing authoritarian messages functioned similarly to primes of friends and family. What specifically in the passages led to this effect? Zmigrod [79] recently suggested that ideological thinking (regardless of its content) possesses two essential structural elements, dogmatism and social connections. Each of these features may help to explain the effects of authoritarian messages on meaning in life.

## Dogmatism/One right answer

With regard to Zmigrod's first feature of ideological thought, dogmatism, a key characteristic of authoritarian views is that they provide a single right answer to complex issues. This aspect of the authoritarian message might be particularly important to its existential appeal. Indeed, expressions of authoritarian values by real-world leaders such as Hitler would suggest that anti-pluralism is one of the hallmark features of these views. For example in his speech at the *Sportpalast* in 1941, he declared, "When 40 or 50 odd parties compete with their gigantic philosophical interests . . . that in itself is a very bad sight; if we only had been rewarded externally for this miserable internal democratic distortion of our lives." One key difference between our egalitarian passage and real-world egalitarian speeches (in Study 2) was that the speeches were persuasive messages (our passage was not). In attempting to persuade, egalitarian speakers may have also focused on conveying a single right answer. This distinction may explain why egalitarian speeches led to lower but not significantly different meaning in life compared to authoritarian speeches.

This line of reasoning suggests that pluralism (valuing the existence of many possible right answers) vs. anti-pluralism may be important to understanding the effects of exposure to authoritarian values on meaning. If dogmatism is key to understanding the effect of authoritarian values on meaning in life, it would suggest that messages conveying single-minded views of other topics, such as American identity, consumer preferences, or religious beliefs, might also enhance meaning in life. Using anti-pluralistic rhetoric to convey pluralistic ideals may also make egalitarian messages more existentially appealing.

If providing one right answer is central to the existential appeal of authoritarian values, then uncertainty may be another causal mechanism to explore. Uncertainty Management Theory [83] suggests people use worldview beliefs to reduce feelings of personal uncertainty, defined as feeling doubtful about oneself and one's beliefs. Exposure to authoritarian values might reduce personal uncertainty, which in turn would enhance the feeling that life is meaningful. This idea is consistent with the Theory of Conservatism as Motivated Social Cognition [48], and Uncertainty Identity Theory [84], which both suggest that ideologies and the groups that endorse them function to reduce uncertainty.

Relatedly, cognitive changes spurred by exposure to authoritarian values and their subsequent effects on mood may help to explain condition effects. Exposure to authoritarian values and the resulting high negative affect and low positive affect, a constellation that produces cognitive inflexibility [85], might enhance cognitive variables that reflect dogmatic thinking, including epistemic beliefs in simple and certain knowledge [86,87] and cognitive closure [49,88]. Exposure to authoritarian values might lead people to perceive less ambiguity in their reality, allowing them to see the world as patterned and predictable, and thereby boost meaning in life.

## Fostering social connection

With regard to Zmigrod's [79] second feature of ideological thought, engendering social connections, exposure to authoritarian values may enhance a sense of belonging to a group. Significance Quest Theory and TMT both place social groups at a focal point of the function of belief systems like right-wing authoritarianism. Consistent with these theories, authoritarian values may enhance meaning in life by making people feel more socially connected, or by engendering a sense of belongingness. Supplemental analyses of Study 2 showed that belongingness partially mediated the effect of condition on meaning in life (See the Supplement pp. 25–26). Authoritarian messages might enhance in-group favoritism and/or out-group hostility, and through these, subsequently affect perceptions of life's meaningfulness. Such considerations implicate race, particularly in the U.S.

Ranging from Black Lives Matter, to gerrymandering and voter suppression, to the controversy over Critical Race Theory, race and racism exist at the heart of many contemporary political issues in the U.S. The positive association between right-wing authoritarianism, particularly authoritarian aggression, and outgroup animus is well-documented [1–6]. The relationship between meaning in life and racism is considerably more nuanced, with past research sometimes showing positive [89], and negative [90] relationships. Given the centrality of race and racism to modern U.S. politics, and the outgroup aggression implicated by authoritarian values, future research should extend the current results by directly assessing their relevance to the effects observed here, and more broadly to racial attitudes and support for policies that advance vs. inhibit equality.

## Moderators of the effect of exposure to authoritarian values on meaning in life

The present studies focused on the main effect of conditions on meaning in life. Future research should probe potential moderators of this effect. It might seem that constructs that are more consistent with the authoritarian message would moderate the effect of condition, including just world beliefs, the belief that positive and negative outcomes are fair and deserved [91], or Protestant Work Ethic, the valuing of hard work, individual achievement, and discipline [92]. Although it is intuitively appealing to imagine that encounters with messages with which one readily agrees would boost meaning in life, this pattern did not emerge in the present studies. Agreement was largely irrelevant to the effect of messages on meaning in life. Analyses probing the concept of existential vacuum (presented in the Supplement) show meaning in life was most strongly boosted among those low on other sources of meaning in life, consistent with theoretical perspectives [45,46]. To some extent, the patterns of these interactions reflect the correlation between right-wing authoritarianism and meaning in life that inspired this work. Those who already embrace conservative or authoritarian ideologies are likely already high in meaning in life. Those who do not endorse such frameworks have more room to grow in the realm of meaning. This line of reasoning suggests that future research might examine baseline levels of meaning in life as a moderator of condition effects on meaning. Those who are generally high on meaning in life may be inoculated against the existential effects of exposure to authoritarian values.

In addition, identifying people for whom an egalitarian worldview provides a satisfying philosophy of life may be important to fully testing whether such a worldview can serve existential functions. For instance, egalitarian worldviews might hold greater existential appeal for those with strong values for universalism (appreciation and tolerance for all people and nature) and self-direction (valuing of independent thought, creation, and exploration) [93]. Egalitarian values may relate especially to a sense of purpose among members of minoritized groups. Results for egalitarian speeches suggest that egalitarian values can be conveyed in ways that do not take a toll on meaning in life.

## Limitations

Limitations of the present studies warrant discussion. First, effect sizes in the current studies were small. However, they are not distinctively small in terms of typical findings in psychological research, and effects of such magnitude on single occasions are often meaningful, particularly at larger scale [94]. Certainly, we do not claim that these effects explain the appeal of authoritarianism in an all-encompassing way.

In the current studies, the manipulation occurred on a single occasion. We would expect that if exposure to authoritarian values occurred on repeated occasions, paired with other

realistic cues that often accompany them in the real-world, effects would be larger. All studies presented authoritarian values using written text. Although this format may reflect some instances of real-world exposure to authoritarian values, future studies should test whether authoritarian values hold similar appeal when presented using other media. We expect that if these ideas were presented from an authoritative figure using realistic vocal and nonverbal cues via video, or in person, their effect on meaning would be strengthened. Additionally, although exposure to authoritarian values often occurs online, future research could also test these ideas in the lab or in the field. Such methodology would facilitate the examination of behavioral measures to extend the present findings, which relied on self-reports.

The spread of right-wing populism in recent years has not been limited to North America. Participants in these studies were exclusively from the United States and Canada. Additional research should test the existential function of exposure to authoritarian values in other regions of the world to determine whether they serve a similar existential function. One possibility, consistent with TMT, may be that authoritarian messages only enhance meaning in life in cultural contexts that are relatively conservative.

The current studies relied on samples recruited from Amazon Mechanical Turk, which carry with them concerns about data quality and naiveté. Given the consistency of results across various operationalizations involving various levels of complexity in 5 separate studies, these do not present strong concerns for the present research. However, future research should replicate these findings in more diverse and nationally representative samples, to address the generalizability of these patterns.

## Conclusion

The emerging popularity of authoritarian values may partially account for the influence of authoritarian values on meaning in life. Even so, our conclusion is no less important. If a popular worldview is one that involves submitting to a powerful leader and ridding society of those who disagree, it is remarkable and disturbing that exposure to this worldview, either through items measuring the construct, historical speeches, or written passages, promotes meaning in life. Frankl (1984) recognized the experience of meaning as the central motivation in human life and that the existential vacuum is experienced as an inner emptiness. Longitudinal designs are necessary to test the possibility that meaning in life, itself, might serve as a buffer against the appeal of authoritarian values: They cannot fill up a life already full of meaning. While these results may be unsettling, in a time when sensitive but vital topics are often avoided in the published literature [95], and when right-wing authoritarian movements have strong mainstream appeal, the present results are crucial to understanding the function of authoritarianism.

## Supporting information

**S1 File. Supplemental materials for all studies.**
(DOCX)

## Acknowledgments

Study 2 was completed in partial fulfillment of an undergraduate Honors Capstone by Sam Luzzo and was presented in a poster at the 2019 Midwestern Psychological Association Convention. The authors thank Christopher S. Sanders for help with the R code used to create the density plots shown in the Supplement, and for providing feedback on the manuscript. The authors thank the anonymous reviewers for their important contribution to this work.

## Author Contributions

**Conceptualization:** Jake Womick, John Eckelkamp, Laura A. King.

**Formal analysis:** Jake Womick, Laura A. King.

**Investigation:** Jake Womick.

**Methodology:** Jake Womick, Sam Luzzo, S. Glenn Baker, Alison Salamun, Laura A. King.

**Project administration:** Jake Womick, John Eckelkamp, Sam Luzzo.

**Resources:** Laura A. King.

**Supervision:** Jake Womick, Laura A. King.

**Visualization:** Jake Womick.

**Writing – original draft:** Jake Womick, Laura A. King.

**Writing – review & editing:** Jake Womick, Sarah J. Ward, S. Glenn Baker, Laura A. King.

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
