## [Decision Letter · Decision Letter 0]

6 Mar 2021

PONE-D-21-00255

Exposure to Authoritarian Values Leads to Lower Positive Affect, Higher Negative Affect, and Higher Meaning in Life

PLOS ONE

Dear Dr. Womick,

Thank you for submitting your manuscript to PLOS ONE. After careful consideration, we feel that it has merit but does not fully meet PLOS ONE’s publication criteria as it currently stands. As you'll see, the reviewers are quite positive about this study, providing as it does a novel set of findings regarding authoritarianism.  At the same time, however, the reviewers raised a number of issues that need to be addressed in a revision. Therefore, we invite you to submit a revised version of the manuscript that addresses the points raised during the review process. Below I summarize some of the main concerns which would need to be addressed.  In addition, each of the reviewers' specific comments needs be addressed (or an explanation provided for not doing so).

A major limitation seems to be the lack of specificity regarding the mechanism(s) underlying the effect you are reporting (noted in one way or another by all reviewers), as well as the possibility that the underlying mechanism(s) might vary over individuals.  Specifically, how and why does reading a message (or responding to a questionnaire) comprised of authoritarian values increase one’s meaning in life? Does message threat somehow increase perceived MIL?  Or is it the suggested solutions to the threat such as increased group cohesion that increases MIL? It is not necessary that you provide an airtight explanation with corresponding data, but it is necessary that you articulate more clearly how this effect occurs.

A second major (and related to the above) limitation is the lack of conceptual clarity regarding the major constructs that you are investigating. Take Meaning in Life for example. Obviously this construct has multiple definitions, components, etc.  But what exactly are you referencing with the term?  The primary dimension you measure is the presence of meaning, but what exactly is that? This is fairly critical because most conceptualizations of meaning seem to reference a construct that is relatively stable.  But what you are demonstrating here is something that is much more fleeting, something that has state-like qualities.  Further explication of the MIL construct is required, with some attention paid to the difference between meaning presence and meaning search (readers may conflate the two, and clearly your findings are relevant only for the former and not the latter).

Similarly, the intended meaning of RWA needs some clarification, and perhaps more importantly, clarification regarding the manner in which it is related to other constructs such as SDO, nationalism, etc.  For example, as one reviewer notes, the extent to which the observed effects are due uniquely to RWA (rather than say SDO or Nationalism) is unclear. The issue of LWA might merit some discussion as well; i.e., would you predict the same patter with LWA as you do with RWA?

The reviewers raised a number of statistical issues (e.g., use of d as an effect size when there are more than 2 groups;  homogeneity of slopes for the ANCOVAs, reporting tests for covariates, etc.) that will need to be addressed.

The use of participants recruited from MTurk is not problematic by itself.  However, there are certain limitations to the use of participants recruited from that platform (e.g., carelessness, lack of naivete, etc.) and these limitations should be noted in the Discussion section. In addition, the compensation paid to your participants appears to be relatively low.  This requires some clarification (e.g., was it roughly equal to minimum wage?) and justification. Also, it appears that there were no attention checks and no participants were excluded from the analyses, a circumstance that seems a little unusual given the relatively large sample.  This should be addressed (e.g., if no Ps were excluded simply state that).

More attention should be paid to the geographic limitations of your research (noted by all reviewers but see especially Reviewer 2).  You are not required to conduct a replication with a non-Western sample (although that would certainly be welcome), but you do need to go to greater (and earlier) lengths to describe the limits to generalizability with the present samples. 

There are a number of typos in the manuscript and it needs a thorough proofreading. Note also that there seems to be some duplication in the SI (e.g., Table 1 in the text and supplementary analyses in the SI for study 1 appear to be the same) and so that needs to be attended to as well.  

Finally, please add a sentence to the manuscript stating where readers can access the data.

In short, I would be happy to consider a revision of your manuscript if you believe you can successfully address these issues as well as the other issues noted by the reviewers.

We look forward to receiving your revised manuscript.

Kind regards,

Thomas Holtgraves, Ph.D.

Academic Editor

PLOS ONE

Journal Requirements:

Reviewers' comments:

Reviewer's Responses to Questions

**Comments to the Author**

1. Is the manuscript technically sound, and do the data support the conclusions?

Reviewer #1: Partly

Reviewer #2: Partly

Reviewer #3: Yes

Reviewer #4: Partly

2. Has the statistical analysis been performed appropriately and rigorously? 

Reviewer #1: I Don't Know

Reviewer #2: Yes

Reviewer #3: Yes

Reviewer #4: I Don't Know

3. Have the authors made all data underlying the findings in their manuscript fully available?

Reviewer #1: Yes

Reviewer #2: Yes

Reviewer #3: Yes

Reviewer #4: Yes

4. Is the manuscript presented in an intelligible fashion and written in standard English?

Reviewer #1: Yes

Reviewer #2: Yes

Reviewer #3: Yes

Reviewer #4: Yes

5. Review Comments to the Author

Reviewer #1: PLOS ONE Ms. No. PONE-D-21-00255: Exposure to Authoritarian Values Leads to Lower Positive Affect, Higher Negative Affect, and Higher Meaning in Life

The authors conduct a series of studies to test the hypothesis that exposure to authoritarian texts leads people to experience more negative affect but more meaning in life. Although the authors’ data support these hypotheses, their manuscript presents some conceptual and statistical issues:

In general, the manuscript needs to be proofread more closely; it contains a number of typographical errors.

The opening section of the introduction seems to be somewhat disjointed, as noted below, although the rest of the introduction is clearer.

p. 3: Discussion of origins of RWA does not address Duckitt’s theory or other relevant theories, although if RWA does have a “positive influence on something that is valuable to people,” that influence probably helps to maintain authoritarian attitudes.

p. 3: Informal observations are not scientific evidence. Is there any research associating RWA with mood or emotions?

p. 3: Why is it “unlikely that listening to or reading about authoritarian values engenders positive feelings, generally”? It seems to me that if one had an authoritarian mindset, one would find reading about our listening to authoritarian values to be pleasing because doing so provides social support for one’s beliefs.

p. 3: Why would exposure to authoritarian statements increase meaning in life? This may be true for people who hold authoritarian values, but why would it apply to people who are not authoritarian?

p. 5: When the authors write “these variables have been separated into different types of well-being” do they mean that the variables have been differentially associated with the two types of well-being? That would seem to make more sense in this context.

p. 5: The authors write “Consistent with past theory [30] and research [31, 32] we expected that the appeal of messages conveying authoritarian values would be existential (rather than affective).” I think that the authors needs to explicate that theory and research a bit: I don’t see the connection based on what they have written.

p. 6: The authors write: “We argue that right-wing authoritarianism is not centrally about making people happy ....” Has anyone argued that RWA is about making people happy? If not, I don’t see the relevance of the authors’ statement.

pp. 9-10: The authors here refer to their hypotheses as counterintuitive. However, on pp. 6-8 they provided both empirical (Womack et al. [although this does not seem to be in the reference list]) and theoretical rationales for their hypotheses.

p. 10: It would be helpful if the authors explicitly stated their hypotheses and recapitulated the theoretical and empirical bases for each hypothesis. For example, the basis of the mood hypothesis in Study 2 (p. 14) was not clear to me.

p. 11 and elsewhere: This is a bit nitpickish, but the authors write “Participants were 68.3% women,” implying that each participants was 68.3% woman and 31.7% man.

p. 13: Did the authors test the assumption of equality of slopes for the covariate across conditions? The same question arises for the other studies.

p. 20: The authors write: “the speech excerpt from Hitler led to the highest meaning in life” but they do not report a statistical test showing that it higher than any of the other authoritarian passages. I think that such a test is needed to support their statement. However, although the Hitler analyses are interesting, they do not seem to central to the goals of the authors’ research and so I think they should be put in the supplementary materials with a footnote directing readers there.

p. 21, Table 2 (and elsewhere): The d statistic is designed to describe the difference between two conditions of an independent variable. How did the authors calculate it for a five-condition analysis? I think that a percentage-of-variance-account-for statistic (such as eta-squared) is more commonly used in this situation.

pp. 21, 23: I assume that the control variable was entered as a covariate. The authors should report the covariate effect as they did for Study 1 (p. 13).

p. 23: What follow-up tests did the authors use to compare conditions of the independent variable? By referring to Bonferroni comparisons, they imply post-hoc tests, but they had a priori hypotheses and so should have conducted a priori tests.

General comment on Studies 2-5: I just got to p. 33, which addresses the question that I have here; however, because this question occurred to me here, perhaps the authors should bring this point up earlier than they do, perhaps in the introduction: It seems to me that one would expect that the effect of authoritarian messages on at least meaning in life would be strongest for people high in authoritarianism because the speeches would help affirm their worldviews. On the other hand, those messages might lead egalitarians to mentally counter-argue, thus also increasing meaning in life. Similarly, authoritarian messages may increase negative affect in authoritarians because those messages increase the salience of their disaffection with modern society and increase egalitarians’ negative affect because the messages offend their values. That is, authoritarian messages have similar effects on authoritarians and egalitarians, but for different reasons. Is there any way to test these possibilities?

p. 24: The authors write “meaning in life resulting from exposure to authoritarian values could predict subsequent positive evaluations of such messages.” This implies a longitudinal approach: current exposure predicts future positive evaluations. However, the studies reported were all cross-sectional.

p. 26, Table 3: (1) The table’s title says that it will include mood, but it only reports message evaluation. (2) I think that “n’s” should be “n =”.

pp. 30-31: The authors are testing a mediating variable hypothesis, not a moderator variable hypothesis: Manipulated conditions —> mood and meaning —> evaluations. Therefore, their moderated regression analysis does not test their hypothesis.

p. 36: The authors write “we speculate that, over time, repeated exposure to authoritarian values, followed by small boosts to meaning, and more favorable evaluations of them, eventually lead people to adopt right-wing authoritarianism.” Given the qualifications at the end of the paragraph, it might be better to say “lead some people” and then say which people in the next sentence.

p. 37: The authors write “Authoritarian values clearly do not fall under the umbrella of virtuous sources of eudaimonic well-being, such as self-actualization and prosocial behavior,” but might authoritarians see it differently? That is, might they not see their values as prosocial in that acceptance of those values would lead to a better society? This idea that authoritarian values lead to better society seems to be in many of the items on the RWA scale.

p. 39: Has there been any research on the complexity of messages in authoritarian versus egalitarian writings and speeches?

p. 39: The Hitler quotation needs a page number as part of the citation.

Reviewer #2: There are a number of issues with this paper that preclude it from publication in its present form.

i) The authors provide the reader with a series of studies done in the north American context (primarily the USA, and one study with a small sample having been conducted in Canada). Essentially, this positions the series of studies within a particular cultural context, yet the vast literature on cultural psychology (of which many proponents are north American) are omitted from the theoretical discussion, experimental design, analyses, and discussions. It is far too problematic to assume these studies have any kind of applicability or insight outside the geographical region in which they are conducted given the current presentation of the paper.

ii) The authors make a general comment about ‘future work’ in other contexts in the discussion. If the authors are genuinely interested in having diverse perspectives on psychological science, then perhaps they could run the study in at least one other cultural context to see what the scopes and limits of their north American findings are.

iii) In study 2, and related discussions in the article, the authors assume that egalitarianism and pluralism are the same or similar. In fact, there is abundant evidence from across the social sciences showing egalitarian values (equality, similarity, etc) are often at odds with pluralistic values (space for diverse cultural values and moral worldviews).

iv) Following on from this, it is unclear from the “egalitarian/pluralistic” narrative on p.16 what the implications are for those who read it? If I am an American liberal participant, and I read this passage, does this mean I should take the moral perspective of American conservatives? The last two points reflect weak theorizing and conceptualization of what pluralism and egalitarianism are and how this conflation might be (mis)understand by research participants.

v) On the topic of egalitarianism, in several studies the authors pay $0.15 to an array of MTurk workers. Can the authors make clear the time it took for workers to do these surveys and whether they are paid under or over the minimum wage?

vi) If one buys in to the validity of these forms of online experimental studies as saying anything important about the actual world (see next point), then it would be more convincing (as well as having non-north American samples) to have a random nationally representative sample within the USA as a further study.

vii) In the introduction the authors briefly discuss observations of authoritarian rallies. Why can’t the authors systematically observe a sample of recordings from these rallies (if attending them and reporting ethnographic observations was not an option?). This would make for an integration of methods where the potential ‘counter-intuitive’ findings could be contextualized and understood in a more holistic – grounded – manner.

viii) Right wing authoritarianism, meaning in life, and affect/mood are not adequately defined in the article. The ontological foundations of these (traits, concepts, ideologies, ‘scales,’?) are not discussion. Are they ‘things’ or are they ‘processes?’ If they are things, why are they things? And I suppose one could justify the broader experimental method based on an ontological conceptualization of RWA, MIL, Mood, as things. But if they are processes (which I suspect they actually are), then the experimental method in its current deployment has very little cogent to say about the nature of the relationship between RWA, MIL, and Mood.

ix) There are numerous smaller issues throughout (e.g. p.3 “the number of attacks by right wing organizations quadrupled”… from what to what? P.4 “life” not “live” etc.)

x) I hope the authors take seriously these comments and aim to improve their manuscript in terms of scientific soundness and ethical appropriateness. In my view the paper will be closer to publication once these flaws are remedied.

Reviewer #3: I enjoyed reading the paper, i found the datasets impressive, the focus novel and interesting and the potential implications of the link betweeen authoritarianism and meaning in life very interesting. That said, I only have some minor comments and i apologise in advance for not giving more constructive feedback, but the comments refer mainly to two overall impressions left with me after reading the paper which i dont think help do the paper justice in its potential contribution and value; firstly i felt the theoretical links / framings were less clear and interconnected, with selected theories being introduced and then used differentially in the discussion to link findings. it would be useful to consider if at all any one framework offers a better alignment overall with the findings of the data, to anchor the findings more thoroughly in theory. secondly, the language of the paper overall , especially in the discussion, is less convincing, with a lot of speculation about what might/may/could potentially be the case here. while i realize the authors are exploring the different possibilities that ground their findings, i do suggest having a read through to identify where the language can be clear and direct, and where it could be framed as more of a speculation around what may/might be the case. these two things coupled together slightly undermine the value of the findings because it reads as if the authors arent fully sure of why the findings are important. lastly, proofread, some minor spelling errors/incomplete sentences, including one in the abstract (' Both studies showed that although egalitarian messages led to better mood and authoritarian messages led to higher MIL. - sentence ends here, either remove 'although' or complete).

Reviewer #4: Peer Review of

Exposure to Authoritarian Values Leads to Lower Positive Affect, Higher Negative

Affect, and Higher Meaning in Life

PLoS One

Spring 2021

I. Summary

The current 5 studies aimed to experimentally test whether exposure to right-wing authoritarian (RWA) attitudes increased negative affect, but concurrently increased self-reported meaning in life (MIL). I must commend the authors on the amount of work they devoted to these 5 studies and the 62-page supplement! I am torn on this manuscript, as I think the hypotheses are compelling, but the evidence perhaps not as much so. I hope I have sufficiently explained my ambivalence below. Please note that I have tried to double-check the main manuscript and the supplementary material (all 124 pages) in case you already addressed some of my concerns; I apologize if I overlooked any of these circumstances.

II. Major Concerns

I have two major concerns that thread the entire manuscript. The first is discriminating between RWA and other similar constructs, particularly SDO (Duckitt & Sibley’s dual process model explicitly integrates the two) and nationalism/patriotism (see Osbourne et al., 2017). Neither of these constructs are directly mentioned in the main manuscript, although pilot studies did examine how much participants thought the speeches’ authors would endorse SDO. In Table S4, the “authoritarian” speeches were rated non-significantly different on SDO and RWA. We find the same pattern in Table S5, except it appears that the authors’ authoritarian speech was rated significantly higher on SDO—but barely. So, are these authoritarian speeches or social dominance-oriented speeches? Although these speeches were not rated for their level of nationalism/patriotism, I would also wonder if we can separate the nationalist vs. authoritarian sentiments from these speeches?

Thus, I question overall whether—or perhaps how much—are the authors uniquely getting at authoritarianism vs. these other constructs? Is it really authoritarianism that serves as an existential cushion or these other constructs? Or a combination?

Further, the authors state “The egalitarian/pluralistic passage was written to offer the opposite perspective of that conveyed in the authoritarian essay” (p. 15). It seems like egalitarianism, which m-w.com defines as “the doctrine that all people are equal and deserve equal rights and opportunities”, would be more of a direct opposite of SDO (whose two components are opposition to equality and group-based dominance), not RWA. Directly opposite attitudes/values would be anti-obedience, anti-conventionalism, and anti-aggression toward those who challenge or deviate the “rules”. I don’t see these values consistently reflected in the egalitarian speeches, especially the authors’ own. (However, I acknowledge that the “egalitarian” essays are not rated significantly higher on SDO than RWA, but I don’t know if that is partly a function of the shortened scales used.)

Relatedly, I have also been intrigued by the construct of left-wing authoritarianism (LWA, essentially support for liberal authoritarianism), which a flurry of research has recently examined, with Lucian Conway leading the charge (Conway et al., 2017). If the authors acknowledge the validity of this construct (and I think recent research has provided some persuasive evidence), would we replicate the same effects we are seeing with right-wing authoritarian/social dominance-oriented/nationalistic messaging?

In any case, Duckitt and Sibley’s dual process model successfully integrates SDO and RWA into a theoretical model. This model maintains that, while they can be strongly correlated, SDO and RWA are distinct constructs that predict different outcomes (e.g., prejudice toward economically competitive vs. socially deviant target groups, respectively; Duckitt, 2006).

So, if we think about alt-right or right-wing rallies, there is some growing consensus that racial animosity/resentment is driving people to these rallies and voting for right-wing candidates (see Warf, 2020). That is, in the face of growing numbers of racial, ethnic (and perhaps sexual and gender minorities), it is the decline of White/cis/hetero power that is scaring the bejesus out of many people, maybe more explicitly among conservatives than liberals.

As the authors discuss “the importance of group identity and group-related feelings” on p. 38 in the GD, I thought “White” (or cis or hetero) seems more akin to group identity and culture than RWA. People can strongly identify with their Whiteness or heterosexuality and talk about protecting White/straight heritage and culture. But I do not think there are a lot of folks going around identifying as an “authoritarian” or discussing “authoritarian” culture, even if to some degree they are endorsing such attitudes or if such attitudes correspond to traditionally “White” or hetero culture.

And perhaps this may be why the “authoritarian” speeches produced such small or very small effects on MIL (note that the effects of these speeches on mood are more persuasive given the consistently moderate ds). Raving about protecting White culture more specifically from immigrants or racial/ethnic minorities may be more existentially fulfilling, while increasing negative affect/lowering positive affect.

But let’s return to the topic of effect sizes, my second major concern. The authors did not address effect sizes for MIL in their General Discussion at all, which was a bit conspicuous. The authors did say that the effect of completing the RWA measure on MIL in S1 was “small” on p. 13, perhaps due to not controlling for mood or not having a more ecologically valid manipulation; but, the effects continued to remain rather small even with the new manipulations. Indeed, the authors admitted as much on p. 24 in discussing the Common Method for S3-S5 (“the effect of interest was likely to be relatively small”) and, the mini-meta-analysis did show an overall unadjusted d = 0.13, with the adjusted d a bit higher (d = 0.28). Cohen’s guidelines indicate that any d < |0.20| is trivial.

That is, if you need over 1,000 people to detect an effect, is it really that important? Is this an efficient use of limited resources? Of course, there are cases where very small effects are truly meaningful. But is that the case here? These are pressing questions that behavioral scientists continue to wrestle with. Given the size of these effects, I think the authors may need to be a bit more tempered in their conclusion and implications in their DG, too.

I am also a bit flummoxed why the adjusted mean differences in MIL, in controlling for mood, are noticeably bigger than the raw mean differences. I have taught statistics > 20 times and analyzed hundreds of datasets using various techniques, but never actually encountered a situation in which the adjusted mean differences were so much larger due to the covariate in an ANCOVA model (I assume the authors conducted ANCOVAs?). I even reached out to three colleagues about this situation, and they were a bit flummoxed, too. Interpreting adjusted means are not always straightforward to begin with (Clason & Mundfrom, 2012).

My colleagues are I speculated that perhaps there was a dramatic change in df between the ANOVA vs. ANCOVA effects. Table 4 does show some differences in df, especially in S4 (btw, the df-between, 1598, for the ANOVA in S5 must be incorrect), but this does not seem to explain this phenomenon. We also considered violations of the homogeneity of slopes assumption, but that assumption was generally met (except in the pilot and S3). My colleagues and I also looked at the pattern of adjusted means; that is, the raw and adjusted Ms in the control condition did not seem that different; however, the adjusted Ms in the Egalitarian condition looked substantially lower, while the adjusted Ms in the Authoritarian condition looked substantially higher. Why are we seeing such shifts in the adjusted Ms in these two conditions, but not the control?

Ultimately, the question is whether the bigger adjusted M differences reflects something meaningful or is a statistical artifice. If it is something meaningful, this may help the authors’ case a bit as we see slightly higher (but still consistently small) ds with the adjusted Ms. But I could not find anything in the manuscript that discussed this phenomenon. Maybe it’s a simple answer; I am very curious to find out.

Additionally, throughout the manuscript I kept thinking that we would only find an increase in MIL after “authoritarian” messaging among conservatives; but, Figure S10 actually showed increased MIL among those low and high in conservatism. This was surprising to me, especially given that Greenberg et al. (1992) found that not only conservatives reported greater dislike of dissimilar others after mortality salience, but liberals actually reported lower dislike after mortality salience. How do the authors reconcile Greenberg et al.’s (1992) results?

Lastly, I am also curious as to what would happen if participants actually knew who wrote each of the real speeches in S2? If people saw Hitler’s or Un’s name, would we still see the increase in MIL? Or would reading a Hitler speech actually alienate people (e.g., decrease belongingness) and decrease MIL?

III. More minor Concerns

1. I would be careful about language. E.g., “Right-wing authoritarianism played a significant role in the 2016 U.S. presidential election” (p. 3); this study showed that RWA and SDO predicted voting intentions, and the study was cross-sectional and not longitudinal, thereby limiting our ability to establish causation. Finding multiple studies, especially longitudinal ones, would better support such a statement.

Another example: “Right-wing authoritarianism involves anti-democratic and anti-social impulses, rendering their appeal baffling to those who do not endorse them”. (p. 5) This sounds a bit politically condescending, perhaps? A liberals’ belief system may seem just as baffling to conversatives’ and vice versa.

2. What do the authors mean by “Environmental coherence” (p. 6)?

3. “Instead, left-wing ideology is positively related to dispositional positive affect” (p. 9); but, it may be important to note that conservatism was related not to higher pos affect but lower neg affect in that study.

4. Nix “definitively” (p. 9), as it sounds too much like “prove”. Also, definitively seems to contradict the strength of the effects found.

5. Missing MTurk payment in S1.

6. S3-S5: The authors paid participants $0.15 vs. $1.00 in S2? Why so little? I feel it is becoming more imperative for us as researchers to pay MTurk participants fairly (indeed, Prolific institutes a min hourly wage), and if the payment does not seem fair, perhaps authors should note it or should address why.

7. I would also be concerned with non-purposeful or careless responding when the payment is so low. The authors mention “exclusions” on p. 10, but I could not find details on these exclusions in the main manuscript or the supplemental .doc. Did the authors screen data for missing responses, completion time, attention checks, etc.? This info should be included in the main paper, IMO.

8. Please put spaces in b/w symbols, operators, and #s. Reading “M(SD)=4.98(1.36), α=.92” is more difficult than “M (SD) = 4.98 (1.36), α = .92.” This aligns with APA style, too (https://www.statisticshowto.com/reporting-statistics-apa-style/).

9. Figure 1: I may recommend altering so that pos and neg affect mean bars are grouped together; the figure now makes it seem like we are supposed to be comparing pos and neg effect for/within each speech, when the authors are most concerned with comparing pos affect between speeches and then neg affect between speeches.

10. I appreciated the discussion of potential mediators in the GD; however, I think the authors need to think how these mediators would be tied to the effects on mood found in the current studies. i.e., Would any of these “cognitive changes” be linked to mood changes (higher neg or lower pos affect) after exposure to “authoritarian” messaging?

 

References (not cited in manuscript)

Clason & Mundfrom, 2012: http://www.glmj.org/archives/articles/Clason_v38n1.pdf

Conway et al., 2017: https://www.researchgate.net/publication/322031834_Finding_the_Loch_Ness_Monster_Left-Wing_Authoritarianism_in_the_United_States

Duckitt, 2006: https://journals.sagepub.com/doi/abs/10.1177/0146167205284282?

Osbourne et al., 2017: https://pubmed.ncbi.nlm.nih.gov/28903711/

Warf, 2020: https://www.routledge.com/Political-Landscapes-of-Donald-Trump/Warf/p/book/9780367197001

6. PLOS authors have the option to publish the peer review history of their article (what does this mean?). If published, this will include your full peer review and any attached files.

Reviewer #1: No

Reviewer #2: No

Reviewer #3: No

Reviewer #4: No

---

## [Author Response · Author response to Decision Letter 0]

28 Jun 2021

Dear Dr. Holtgraves,

Thank you for the opportunity to revise our submission (PONE-D-21-00255, “Exposure to Authoritarian Values Leads to Lower Positive Affect, Higher Negative Affect, and Higher Meaning in Life”). We have now completed the changes to the paper in response to your thoughtful feedback and the reviewers’ critiques. Below we explain how we have responded to these. 

Editor’s Feedback

You posed several questions that we responded to individually:

1. A major limitation seems to be the lack of specificity regarding the mechanism(s) underlying the effect you are reporting (noted in one way or another by all reviewers), as well as the possibility that the underlying mechanism(s) might vary over individuals. Specifically, how and why does reading a message (or responding to a questionnaire) comprised of authoritarian values increase one’s meaning in life? Does message threat somehow increase perceived MIL? Or is it the suggested solutions to the threat such as increased group cohesion that increases MIL? It is not necessary that you provide an airtight explanation with corresponding data, but it is necessary that you articulate more clearly how this effect occurs.

Response: Thank you for highlighting this issue. In response to this and numerous other reviewer comments, we have conducted a number of revisions that address these concerns. First, we have conducted additional analyses addressing whether potential underlying mechanisms may vary over individuals. Although we view this as a compelling possibility, our data did not support it in the case of dispositional conservative ideology and affect (see response to reviewer 1 comment 18). With regard to message threat, our data again do not support this as a compelling explanation—perceptions of moral superiority, anger, negative affect, and disagreement did not interact with condition to predict meaning in a way that would suggest these serve as explanations (see Supplement pp. 56-65), at least at the conscious level. Of the possibilities you raise, the idea that exposure to authoritarian values affects group-level variables is perhaps most compelling. Indeed, we found belongingness partially mediates the effect of condition on meaning in life (Supplement p. 25). Compelled by your feedback, we have additionally revised the general discussion to focus more clearly on these issues (pp 41-44), framing our reasoning around a recent paper on two key aspects of ideological thinking, dogmatism and social relatedness (Zmigrod, 2021). 

Conceptual Ambiguities

2. A second major (and related to the above) limitation is the lack of conceptual clarity regarding the major constructs that you are investigating. Take Meaning in Life for example. Obviously this construct has multiple definitions, components, etc. But what exactly are you referencing with the term? The primary dimension you measure is the presence of meaning, but what exactly is that? This is fairly critical because most conceptualizations of meaning seem to reference a construct that is relatively stable. But what you are demonstrating here is something that is much more fleeting, something that has state-like qualities. Further explication of the MIL construct is required, with some attention paid to the difference between meaning presence and meaning search (readers may conflate the two, and clearly your findings are relevant only for the former and not the latter).

Response: We thank you for this feedback and are more than a little embarrassed that we did not ever define mood in the paper. We now provide a formal definition of mood and give examples of positive and negative affect descriptors. We have also replaced the previous definition of meaning in life with one that is more directly related to meaning in life as an aspect of well-being. We have added sample items in introduction so that readers know what we are talking about. Additionally, we explicitly state that, although meaning in life is often defined as arising from three lower-order components, we focus on the global experience of meaning that arises from them in the current research (p. 4). We also review past research in order to inform readers that meaning in life is generally stable over time, but also fluctuates in response to changing environmental circumstances, and experimental manipulations (pp. 4-5). We also discuss the differences between presence and search for meaning in the text, clarifying that we are only interested in the presence of meaning in life (p. 51). Our research questions concern whether exposure to authoritarian values enhance the experience of meaning in life. High scores on search for meaning do not necessarily correspond to experiencing one’s life as meaningful, and low scores do not mean a person experiences life as meaningless. Additionally, the link between presence and search is inconsistent, with past research identifying both positive and negative links (Chu & Fung, 2021). For these reasons, we focus only on presence of meaning as an indicator of global meaning in life. 

3. Similarly, the intended meaning of RWA needs some clarification, and perhaps more importantly, clarification regarding the manner in which it is related to other constructs such as SDO, nationalism, etc. For example, as one reviewer notes, the extent to which the observed effects are due uniquely to RWA (rather than say SDO or Nationalism) is unclear. The issue of LWA might merit some discussion as well; i.e., would you predict the same patter with LWA as you do with RWA?

Response: We now address SDO in the introduction, clarifying that this construct is not related to meaning in life (p. 6). We also have added discussion of LWA to the text (p. 40). Please also see response to reviewer 4 comments 1 and 4. 

4. The reviewers raised a number of statistical issues (e.g., use of d as an effect size when there are more than 2 groups; homogeneity of slopes for the ANCOVAs, reporting tests for covariates, etc.) that will need to be addressed.

Response: Please see responses to reviewers 1 (comments 15-17) and 4 (comments 6 and 7) below. 

5. The use of participants recruited from MTurk is not problematic by itself. However, there are certain limitations to the use of participants recruited from that platform (e.g., carelessness, lack of naivete, etc.) and these limitations should be noted in the Discussion section. In addition, the compensation paid to your participants appears to be relatively low. This requires some clarification (e.g., was it roughly equal to minimum wage?) and justification. Also, it appears that there were no attention checks and no participants were excluded from the analyses, a circumstance that seems a little unusual given the relatively large sample. This should be addressed (e.g., if no Ps were excluded simply state that).

Response: Thank you for highlighting these issues. We have added discussion of limitations associated with MTurk data to the text (pp. 47-48). All of these Studies were very brief, and we have been working on data collection for quite some time, so pay standards have changed over the course of administering them. We have noted these details in the methods for each Study. Additionally, we did not exclude participants from analyses, and now state that in the text, p. 12. 

6. More attention should be paid to the geographic limitations of your research (noted by all reviewers but see especially Reviewer 2). You are not required to conduct a replication with a non-Western sample (although that would certainly be welcome), but you do need to go to greater (and earlier) lengths to describe the limits to generalizability with the present samples. 

Response: We now note in the opening paragraphs and general discussion that our data were limited to U.S. (and Canada) and consider how these limit the generalizability of our results. 

7. There are a number of typos in the manuscript and it needs a thorough proofreading. Note also that there seems to be some duplication in the SI (e.g., Table 1 in the text and supplementary analyses in the SI for study 1 appear to be the same) and so that needs to be attended to as well. 

Response: Thank you for paying such close attention to our work. We have conducted a thorough proofreading, addressing a number of errors, and have removed the Study 1 analyses from the Supplement.

8. Finally, please add a sentence to the manuscript stating where readers can access the data.

 Response: A data availability statement is on p. 48, and we have also added it to p. 12. 

Reviewer #1

The authors conduct a series of studies to test the hypothesis that exposure to authoritarian texts leads people to experience more negative affect but more meaning in life. Although the authors’ data support these hypotheses, their manuscript presents some conceptual and statistical issues:

1. In general, the manuscript needs to be proofread more closely; it contains a number of typographical errors.

Response: Thank you for your close attention to our work. We have conducted a thorough proofreading. 

2. The opening section of the introduction seems to be somewhat disjointed, as noted below, although the rest of the introduction is clearer.

Response: We have revised the opening section to be more cohesive. 

3. p. 3: Discussion of origins of RWA does not address Duckitt’s theory or other relevant theories, although if RWA does have a “positive influence on something that is valuable to people,” that influence probably helps to maintain authoritarian attitudes.

Response: We now incorporate Duckitt’s Dual process theory in the introduction (p. 6) and general discussion (p. 40), and note that we do not view the existential function of authoritarian values as the only factor that potentially draws people to these views (p. 47). 

4. p. 3: Informal observations are not scientific evidence. Is there any research associating RWA with mood or emotions?

Response: We cut this and now only cite research. 

5. p. 3: Why is it “unlikely that listening to or reading about authoritarian values engenders positive feelings, generally”? It seems to me that if one had an authoritarian mindset, one would find reading about our listening to authoritarian values to be pleasing because doing so provides social support for one’s beliefs.

Response: We removed this from the manuscript. 

6. p. 3: Why would exposure to authoritarian statements increase meaning in life? This may be true for people who hold authoritarian values, but why would it apply to people who are not authoritarian?

Response: Although mediation is not the focus of this paper, we expect, based on correlational research showing a positive link between right-wing authoritarianism and meaning in life, that exposing people to expressions of these views, which characterize the world in certain and unambiguous terms, with clear cut distinctions between groups, that they momentarily enhance perceptions of meaning (p. 9). In contrast to your reasoning, most theoretical views would in fact suggest the opposite—that these would be especially meaningful for people who are not authoritarian because they do not already endorse this dispositional source of meaning in life. 

7. p. 5: When the authors write “these variables have been separated into different types of well-being” do they mean that the variables have been differentially associated with the two types of well-being? That would seem to make more sense in this context.

Response: We cut the section on hedonic vs. eudaimonic well-being. 

8. p. 5: The authors write “Consistent with past theory [30] and research [31, 32] we expected that the appeal of messages conveying authoritarian values would be existential (rather than affective).” I think that the authors needs to explicate that theory and research a bit: I don’t see the connection based on what they have written.

Response: The reasoning is provided later. We have cut this statement. 

9. p. 6: The authors write: “We argue that right-wing authoritarianism is not centrally about making people happy ....” Has anyone argued that RWA is about making people happy? If not, I don’t see the relevance of the authors’ statement.

Response: We removed this statement from the text.. 

10. pp. 9-10: The authors here refer to their hypotheses as counterintuitive. However, on pp. 6-8 they provided both empirical (Womack et al. [although this does not seem to be in the reference list]) and theoretical rationales for their hypotheses.

Response: We no longer refer to our predictions as counterintuitive. The citation was in the reference list—the author’s name is Womick.

11. p. 10: It would be helpful if the authors explicitly stated their hypotheses and recapitulated the theoretical and empirical bases for each hypothesis. For example, the basis of the mood hypothesis in Study 2 (p. 14) was not clear to me.

Response: We have added a recapitulation of evidence for our hypotheses, p. 9. 

12. p. 11 and elsewhere: This is a bit nitpickish, but the authors write “Participants were 68.3% women,” implying that each participants was 68.3% woman and 31.7% man.

Response: Thank you for noting this error, we have changed this throughout so that gender refers to the sample not persons. 

13. p. 13: Did the authors test the assumption of equality of slopes for the covariate across conditions? The same question arises for the other studies.

Response: We have added these analyses to the text for Studies 3-5, pp. 32-37. For Study 2, due to the large number of conditions, we determined to simply examine correlations between mood and meaning in life within each condition, rather than formally testing for moderation. Results are shown in the Supplement, p. 24. PA was positively associated with meaning in life in all conditions, and only the original authoritarian and egalitarian conditions significantly differed, z = 1.84, p = . 03. Negative affect was negatively related to meaning in life in all conditions. Although the relationship was non-significant in the original egalitarian condition, this relationship did not significantly differ from those obtained in any other condition. Ultimately, these results did not enhance our understanding of the effect of condition on mood and meaning in life, so we report them in the Supplement. 

14. p. 20: The authors write: “the speech excerpt from Hitler led to the highest meaning in life” but they do not report a statistical test showing that it higher than any of the other authoritarian passages. I think that such a test is needed to support their statement. However, although the Hitler analyses are interesting, they do not seem to central to the goals of the authors’ research and so I think they should be put in the supplementary materials with a footnote directing readers there.

Response: We thank the Reviewer for this suggestion. These analyses have been moved to the Supplement.

15. p. 21, Table 2 (and elsewhere): The d statistic is designed to describe the difference between two conditions of an independent variable. How did the authors calculate it for a five-condition analysis? I think that a percentage-of-variance-account-for statistic (such as eta-squared) is more commonly used in this situation.

Response: Cohen’s d and eta squared can be derived from each other. We have used d here because it is more commonly used and more intuitive for many readers. 

16. pp. 21, 23: I assume that the control variable was entered as a covariate. The authors should report the covariate effect as they did for Study 1 (p. 13). 

Response: We have reported the effects for covariates in all models, pp. 22, and 49.

17. p. 23: What follow-up tests did the authors use to compare conditions of the independent variable? By referring to Bonferroni comparisons, they imply post-hoc tests, but they had a priori hypotheses and so should have conducted a priori tests.

Response: We continue to use Bonferroni comparisons because they are a more conservative test of the differences among the conditions. We feel this is appropriate because our predictions for egalitarianism were not strong. 

18. General comment on Studies 2-5: I just got to p. 33, which addresses the question that I have here; however, because this question occurred to me here, perhaps the authors should bring this point up earlier than they do, perhaps in the introduction: It seems to me that one would expect that the effect of authoritarian messages on at least meaning in life would be strongest for people high in authoritarianism because the speeches would help affirm their worldviews. On the other hand, those messages might lead egalitarians to mentally counter-argue, thus also increasing meaning in life. Similarly, authoritarian messages may increase negative affect in authoritarians because those messages increase the salience of their disaffection with modern society and increase egalitarians’ negative affect because the messages offend their values. That is, authoritarian messages have similar effects on authoritarians and egalitarians, but for different reasons. Is there any way to test these possibilities? 

Response: We do not have the precise data to address these possibilities. We indirectly address them in the text and supplement by testing for moderation by mood, and three-way interactions for anger, disagreement, and condition. Compelled by your comment, we conducted additional analyses. In these, we tested whether a three-way interaction occurred between conservatism, mood, and condition that would suggest among those for whom authoritarianism is worldview consistent, positive mood enhances meaning; and among those for whom it is worldview inconsistent, negative mood enhances meaning. In no case did we find evidence for such a pattern. These models only showed the two-way interactions for PA and conservatism with condition that we already report. However, we also offer the possibility that, because authoritarianism serves as a source of meaning in life (but not mood), it makes sense without further explanation that it enhances meaning in life, and does so particularly among those who are lacking in other sources of meaning (in these cases, dispositional conservative ideology and positive affect). These patterns are consistent with what we expect based on other correlational research, and past experimental research on meaning in life, as we now review in the intro. 

19. p. 24: The authors write “meaning in life resulting from exposure to authoritarian values could predict subsequent positive evaluations of such messages.” This implies a longitudinal approach: current exposure predicts future positive evaluations. However, the studies reported were all cross-sectional.

Response: In the context of experiments when the mediators (here, meaning in life and mood) have been measured prior to the dependent measures (evaluations) our prediction is appropriate—it does not require longitudinal data. 

21. p. 26, Table 3: (1) The table’s title says that it will include mood, but it only reports message evaluation. (2) I think that “n’s” should be “n =”.

Response: We have completed these revisions.

22. pp. 30-31: The authors are testing a mediating variable hypothesis, not a moderator variable hypothesis: Manipulated conditions —> mood and meaning —> evaluations. Therefore, their moderated regression analysis does not test their hypothesis.

Response: The analyses we report essentially provides the same information as that provided but also adds the test for moderation. 

23. p. 36: The authors write “we speculate that, over time, repeated exposure to authoritarian values, followed by small boosts to meaning, and more favorable evaluations of them, eventually lead people to adopt right-wing authoritarianism.” Given the qualifications at the end of the paragraph, it might be better to say “lead some people” and then say which people in the next sentence.

Response: We added “some” on p. 39

24. p. 37: The authors write “Authoritarian values clearly do not fall under the umbrella of virtuous sources of eudaimonic well-being, such as self-actualization and prosocial behavior,” but might authoritarians see it differently? That is, might they not see their values as prosocial in that acceptance of those values would lead to a better society? This idea that authoritarian values lead to better society seems to be in many of the items on the RWA scale.

Response: We have cut the issue of hedonic vs. eudaimonic well-being from the manuscript. 

25. p. 39: Has there been any research on the complexity of messages in authoritarian versus egalitarian writings and speeches?

Response: We assessed the complexity in numerous ways. We assessed the reading-level of the passages we authored, which did not differ in a systematic way that would explain condition effects (Supplement, p. 3). We also asked participants to rate how easy it was to read and understand the passages (Supplement, pp 20 and 22), and these did not explain the effect of condition on meaning in life. In general, right-wing messages tend to show lower integrative complexity, but this difference also depends on the speaker and to whom they are speaking (Houck & Conway, 2019). We did not incorporate this issue into the paper because it is beyond the scope of this work. 

26. p. 39: The Hitler quotation needs a page number as part of the citation.

Response: We note in the text that the quote comes from a speech (at the Sportpalatz in 1941). No page number is available. 

Reviewer #2

i) The authors provide the reader with a series of studies done in the north American context (primarily the USA, and one study with a small sample having been conducted in Canada). Essentially, this positions the series of studies within a particular cultural context, yet the vast literature on cultural psychology (of which many proponents are north American) are omitted from the theoretical discussion, experimental design, analyses, and discussions. It is far too problematic to assume these studies have any kind of applicability or insight outside the geographical region in which they are conducted given the current presentation of the paper.

Response: We now note in the opening paragraphs that these studies focus on the North American cultural context, and discuss limits on generalizability in the General Discussion.

ii) The authors make a general comment about ‘future work’ in other contexts in the discussion. If the authors are genuinely interested in having diverse perspectives on psychological science, then perhaps they could run the study in at least one other cultural context to see what the scopes and limits of their north American findings are.

Response: We now note that future research ought to be conducted outside North America (p. 47). However, we believe the present data are useful and potentially important within the context of the U.S.

iii) In study 2, and related discussions in the article, the authors assume that egalitarianism and pluralism are the same or similar. In fact, there is abundant evidence from across the social sciences showing egalitarian values (equality, similarity, etc) are often at odds with pluralistic values (space for diverse cultural values and moral worldviews).

Response: Although we disagree, we have cut all mentions of egalitarian views being more pluralistic from the text.

iv) Following on from this, it is unclear from the “egalitarian/pluralistic” narrative on p.16 what the implications are for those who read it? If I am an American liberal participant, and I read this passage, does this mean I should take the moral perspective of American conservatives? The last two points reflect weak theorizing and conceptualization of what pluralism and egalitarianism are and how this conflation might be (mis)understand by research participants.

Response: As we note in the text, all participants were instructed, “While reading, try to suspend judgment and just learn about this person’s experience. Try to focus on the ideas conveyed without immediately judging the author.” So, participants were not expected to take on the views of the writer. 

v) On the topic of egalitarianism, in several studies the authors pay $0.15 to an array of MTurk workers. Can the authors make clear the time it took for workers to do these surveys and whether they are paid under or over the minimum wage?

Response: These Studies took, on average, 5 minutes to complete. Although the pay may seem quite low, the effort required was minimal. 

vi) If one buys in to the validity of these forms of online experimental studies as saying anything important about the actual world (see next point), then it would be more convincing (as well as having non-north American samples) to have a random nationally representative sample within the USA as a further study.

Response: Thank you for this insightful comment, we now note this issue in the General Discussion (p. 47). 

vii) In the introduction the authors briefly discuss observations of authoritarian rallies. Why can’t the authors systematically observe a sample of recordings from these rallies (if attending them and reporting ethnographic observations was not an option?). This would make for an integration of methods where the potential ‘counter-intuitive’ findings could be contextualized and understood in a more holistic – grounded – manner.

Response: The reviewer’s suggestions is a really interesting one and one that might well be pursued. However, the observations about the people in the crowds has been cut in response to another reviewer’s comments. 

viii) Right wing authoritarianism, meaning in life, and affect/mood are not adequately defined in the article. The ontological foundations of these (traits, concepts, ideologies, ‘scales,’?) are not discussion. Are they ‘things’ or are they ‘processes?’ If they are things, why are they things? And I suppose one could justify the broader experimental method based on an ontological conceptualization of RWA, MIL, Mood, as things. But if they are processes (which I suspect they actually are), then the experimental method in its current deployment has very little cogent to say about the nature of the relationship between RWA, MIL, and Mood.

Response: Thank you for highlighting these shortcomings. We have provided clearer definitions for these. Like with all psychological constructs, these are likely best thought of as processes that we are assessing in a particular moment in time as “things.” 

ix) There are numerous smaller issues throughout (e.g. p.3 “the number of attacks by right wing organizations quadrupled”… from what to what? P.4 “life” not “live” etc.)

Response: We have added additional and updated information about these on p. 3, and fixed the error on p. 4

Reviewer #3

1. I enjoyed reading the paper…, That said, I only have some minor comments and i apologise in advance for not giving more constructive feedback, but the comments refer mainly to two overall impressions left with me after reading the paper which i dont think help do the paper justice in its potential contribution and value; 

Response: Thank you, we are encouraged by your positive feedback. 

2. firstly i felt the theoretical links / framings were less clear and interconnected, with selected theories being introduced and then used differentially in the discussion to link findings. it would be useful to consider if at all any one framework offers a better alignment overall with the findings of the data, to anchor the findings more thoroughly in theory. 

Response: In this revision we have tried to be more precise with our definitions, the theories on which we draw, the conclusions we reach. 

3. secondly, the language of the paper overall , especially in the discussion, is less convincing, with a lot of speculation about what might/may/could potentially be the case here. 

Response: We have tried to anchor the General Discussion with an eye toward more clear potential mediators and we have cut some of the speculation. 

4. lastly, proofread, some minor spelling errors/incomplete sentences, including one in the abstract (' Both studies showed that although egalitarian messages led to better mood and authoritarian messages led to higher MIL. - sentence ends here, either remove 'although' or complete).

Response: We cut “although.”

Reviewer #4: 

II. Major Concerns

1. I have two major concerns that thread the entire manuscript. The first is discriminating between RWA and other similar constructs, particularly SDO (Duckitt & Sibley’s dual process model explicitly integrates the two) and nationalism/patriotism (see Osbourne et al., 2017). Neither of these constructs are directly mentioned in the main manuscript, although pilot studies did examine how much participants thought the speeches’ authors would endorse SDO. In Table S4, the “authoritarian” speeches were rated non-significantly different on SDO and RWA. We find the same pattern in Table S5, except it appears that the authors’ authoritarian speech was rated significantly higher on SDO—but barely. So, are these authoritarian speeches or social dominance-oriented speeches? Although these speeches were not rated for their level of nationalism/patriotism, I would also wonder if we can separate the nationalist vs. authoritarian sentiments from these speeches?

Response: Although nationalism is outside of the scope of the current paper (also note that Study 1 simply used an RWA scale, and Study 2 used speeches from countries across the globe, including Russia, Germany, and North Korea, so nationalism is an unlikely alternative explanation), we have now dedicated more space in the manuscript to describing the dual process model in both the introduction and general discussion. Social Dominance Orientation does not serve an existential function in the same way that right-wing authoritarianism does (its negatively related to meaning in life), and thus was not considered in the current research, and cannot serve as an alternative explanation. 

2. Thus, I question overall whether—or perhaps how much—are the authors uniquely getting at authoritarianism vs. these other constructs? Is it really authoritarianism that serves as an existential cushion or these other constructs? Or a combination?

Response: The passages in Studies 3-5 consist almost entirely of items used to measure right-wing authoritarianism. And, recall, that in Study 1, simply being exposed to the RWA scale before MIL scale enhanced meaning in life. Considered together, we feel confident our effects indeed are about right-wing authoritarianism. 

3. Further, the authors state “The egalitarian/pluralistic passage was written to offer the opposite perspective of that conveyed in the authoritarian essay” (p. 15). It seems like egalitarianism, which m-w.com defines as “the doctrine that all people are equal and deserve equal rights and opportunities”, would be more of a direct opposite of SDO (whose two components are opposition to equality and group-based dominance), not RWA. Directly opposite attitudes/values would be anti-obedience, anti-conventionalism, and anti-aggression toward those who challenge or deviate the “rules”. I don’t see these values consistently reflected in the egalitarian speeches, especially the authors’ own. (However, I acknowledge that the “egalitarian” essays are not rated significantly higher on SDO than RWA, but I don’t know if that is partly a function of the shortened scales used.)

Response: We believe it is highly unlikely that egalitarian speeches would be rated higher on either of these if we administered the full scales in the pilot study. Although your point is well-taken, that egalitarianism conceptually may more closely mirror social dominance, we maintain that we designed the egalitarian condition to be the opposite of passage expressing authoritarian values. Indeed, the passage addresses aspects of the Merriam-Webster definition you provide. For instance, it addresses that all people are equal, “we must remember to maintain regard for equality, human dignity and freedom;” and expresses value for diversity (rather than conformity and cohesion), “immerse oneself in a diverse group of people that come from different backgrounds and espouse a variety values and moral beliefs.” The passage also addresses anti-obedience, “we should support open-minded leaders that listen to their constituents and act in accordance with our collective interests;” anti-conventionalism, “reinvent the traditions of our country. In my opinion, it is important to think independently. We cannot simply trust the conventions of our society without questioning them and improving them;” and anti-aggression, “In embracing people that are different from ourselves, we can learn from other cultures… maintain regard for equality, human dignity and freedom."

4. Relatedly, I have also been intrigued by the construct of left-wing authoritarianism (LWA, essentially support for liberal authoritarianism), which a flurry of research has recently examined, with Lucian Conway leading the charge (Conway et al., 2017). If the authors acknowledge the validity of this construct (and I think recent research has provided some persuasive evidence), would we replicate the same effects we are seeing with right-wing authoritarian/social dominance-oriented/nationalistic messaging?

Response: We now acknowledge left-wing authoritarianism in the text. However, we do not expect it would show similar effects. A new left-wing authoritarianism scale was recently published in JPSP, and we have unpublished data showing it, like other left-wing constructs, is related negatively to the experience of meaning in life. 

5. In any case, Duckitt and Sibley’s dual process model successfully integrates SDO and RWA into a theoretical model. This model maintains that, while they can be strongly correlated, SDO and RWA are distinct constructs that predict different outcomes (e.g., prejudice toward economically competitive vs. socially deviant target groups, respectively; Duckitt, 2006).

So, if we think about alt-right or right-wing rallies, there is some growing consensus that racial animosity/resentment is driving people to these rallies and voting for right-wing candidates (see Warf, 2020). That is, in the face of growing numbers of racial, ethnic (and perhaps sexual and gender minorities), it is the decline of White/cis/hetero power that is scaring the bejesus out of many people, maybe more explicitly among conservatives than liberals.

As the authors discuss “the importance of group identity and group-related feelings” on p. 38 in the GD, I thought “White” (or cis or hetero) seems more akin to group identity and culture than RWA. People can strongly identify with their Whiteness or heterosexuality and talk about protecting White/straight heritage and culture. But I do not think there are a lot of folks going around identifying as an “authoritarian” or discussing “authoritarian” culture, even if to some degree they are endorsing such attitudes or if such attitudes correspond to traditionally “White” or hetero culture.

And perhaps this may be why the “authoritarian” speeches produced such small or very small effects on MIL (note that the effects of these speeches on mood are more persuasive given the consistently moderate ds). Raving about protecting White culture more specifically from immigrants or racial/ethnic minorities may be more existentially fulfilling, while increasing negative affect/lowering positive affect.

Response: Thank you for sharing these insightful ideas. We have added to the general discussion, p. 42, and certainly hope they are addressed in future research. 

6. But let’s return to the topic of effect sizes, my second major concern. The authors did not address effect sizes for MIL in their General Discussion at all, which was a bit conspicuous. The authors did say that the effect of completing the RWA measure on MIL in S1 was “small” on p. 13, perhaps due to not controlling for mood or not having a more ecologically valid manipulation; but, the effects continued to remain rather small even with the new manipulations. Indeed, the authors admitted as much on p. 24 in discussing the Common Method for S3-S5 (“the effect of interest was likely to be relatively small”) and, the mini-meta-analysis did show an overall unadjusted d = 0.13, with the adjusted d a bit higher (d = 0.28). Cohen’s guidelines indicate that any d < |0.20| is trivial.

Response: We have added consideration of the effect sizes to the GD, pp 45-46. 

7. I am also a bit flummoxed why the adjusted mean differences in MIL, in controlling for mood, are noticeably bigger than the raw mean differences. I have taught statistics > 20 times and analyzed hundreds of datasets using various techniques, but never actually encountered a situation in which the adjusted mean differences were so much larger due to the covariate in an ANCOVA model (I assume the authors conducted ANCOVAs?). I even reached out to three colleagues about this situation, and they were a bit flummoxed, too. Interpreting adjusted means are not always straightforward to begin with (Clason & Mundfrom, 2012).

Ultimately, the question is whether the bigger adjusted M differences reflects something meaningful or is a statistical artifice. If it is something meaningful, this may help the authors’ case a bit as we see slightly higher (but still consistently small) ds with the adjusted Ms. But I could not find anything in the manuscript that discussed this phenomenon. Maybe it’s a simple answer; I am very curious to find out.

Response: In the text, we note the consistent tendency of controlling for mood to lower MIL in the egalitarian condition and increase MIL in the authoritarian condition (pp. 32-37). We now report exploratory analyses probing these issues of slopes.

8. Additionally, throughout the manuscript I kept thinking that we would only find an increase in MIL after “authoritarian” messaging among conservatives; but, Figure S10 actually showed increased MIL among those low and high in conservatism. This was surprising to me, especially given that Greenberg et al. (1992) found that not only conservatives reported greater dislike of dissimilar others after mortality salience, but liberals actually reported lower dislike after mortality salience. How do the authors reconcile Greenberg et al.’s (1992) results?

Response: It is clear that what is occurring in these studies is not identical to a TMT mortality salience effect. Methodologically, it is important to consider that all of their studies employ a 10+ minute delay after mortality salience because they view these worldview defense behaviors as a distal reaction to death anxiety. In comparison, measures of mood and meaning in these studies were collected immediately after the manipulation. Our explanation for this pattern of moderation is essentially that conservatives were at a ceiling for meaning in life. In comparison, liberals, who do not endorse a view that dispositionally promotes meaning in life, experienced a situation in which they were exposed to ideological content that momentarily enhanced perceptions of meaning. Again, why this effect occurs is not the purpose of the current studies. Supplemental analyses show it is at least in part because authoritarian values enhance perceptions of belonging. However, it may also be that exposure to extreme ideological content of any nature (that provides dogmatic descriptions and prescriptions for living, as well as conveys ingroup-favoritism and out-group animus, Zmigrod, 2021) could similarly affect anyone, regardless of their pre-existing views. We now discuss this possibility in more detail in the text. 

9. Lastly, I am also curious as to what would happen if participants actually knew who wrote each of the real speeches in S2? If people saw Hitler’s or Un’s name, would we still see the increase in MIL? Or would reading a Hitler speech actually alienate people (e.g., decrease belongingness) and decrease MIL?

Response: This question is definitely interesting but not something we can broach with the present data. 

III. More minor Concerns

10. I would be careful about language. E.g., “Right-wing authoritarianism played a significant role in the 2016 U.S. presidential election” (p. 3); this study showed that RWA and SDO predicted voting intentions, and the study was cross-sectional and not longitudinal, thereby limiting our ability to establish causation. Finding multiple studies, especially longitudinal ones, would better support such a statement.

Another example: “Right-wing authoritarianism involves anti-democratic and anti-social impulses, rendering their appeal baffling to those who do not endorse them”. (p. 5) This sounds a bit politically condescending, perhaps? A liberals’ belief system may seem just as baffling to conversatives’ and vice versa.

Response: We have added additional citations to p. 3, and have removed the problematic phrase from p. 5. 

11. What do the authors mean by “Environmental coherence” (p. 6)?

Response: We have clarified this means exposure to stimuli that “make sense”. 

3. “Instead, left-wing ideology is positively related to dispositional positive affect” (p. 9); but, it may be important to note that conservatism was related not to higher pos affect but lower neg affect in that study.

Response: We have revised the sentence on p. 10.

4. Nix “definitively” (p. 9), as it sounds too much like “prove”. Also, definitively seems to contradict the strength of the effects found.

Response: We have removed “definitively” from the text. .

5. Missing MTurk payment in S1.

Response: We have added the payment for S1. 

6. S3-S5: The authors paid participants $0.15 vs. $1.00 in S2? Why so little? I feel it is becoming more imperative for us as researchers to pay MTurk participants fairly (indeed, Prolific institutes a min hourly wage), and if the payment does not seem fair, perhaps authors should note it or should address why.

Response: Study 2 took 8 minutes, and Studies 3-5 took 5 minutes. Additionally, we have been working on these studies for roughly 6 years, and pay standards have changed over that time. 

7. I would also be concerned with non-purposeful or careless responding when the payment is so low. The authors mention “exclusions” on p. 10, but I could not find details on these exclusions in the main manuscript or the supplemental .doc. Did the authors screen data for missing responses, completion time, attention checks, etc.? This info should be included in the main paper, IMO.

Response: Yes, those with missing data were automatically excluded from analyses (we used SPSS to analyze the data). We included attention checks, however did not pre-register dropping any participants from analyses. After conducting our pre-registered analyses, we conducted robustness checks, re-running analyses with those who failed the attention checks excluded. Results did not differ when we excluded these inattentive responders (we expect, due to the large sample sizes). Thus, to maintain consistency with our pre-registration, we present results with all participants included, and now note this in the text, p. 12. 

8. Please put spaces in b/w symbols, operators, and #s. Reading “M(SD)=4.98(1.36), α=.92” is more difficult than “M (SD) = 4.98 (1.36), α = .92.” This aligns with APA style, too (https://www.statisticshowto.com/reporting-statistics-apa-style/).

Response: We have added spaces throughout the manuscript.

9. Figure 1: I may recommend altering so that pos and neg affect mean bars are grouped together; the figure now makes it seem like we are supposed to be comparing pos and neg effect for/within each speech, when the authors are most concerned with comparing pos affect between speeches and then neg affect between speeches.

Response: We have revised the figures.

10. I appreciated the discussion of potential mediators in the GD; however, I think the authors need to think how these mediators would be tied to the effects on mood found in the current studies. i.e., Would any of these “cognitive changes” be linked to mood changes (higher neg or lower pos affect) after exposure to “authoritarian” messaging?

Response: Yes, in general, negative mood enhances cognitive inflexibility, and all of these are markers of rigid thinking. Thank you for noting this, and we have now explicitly stated so in the manuscript (p. 44). 

Again, thank you for your thoughtful feedback on our work, and the continued opportunity to have this work considered for publication. 

Sincerely,

Jake Womick

---

## [Editor Report · Decision Letter 1]

15 Jul 2021

PONE-D-21-00255R1

Exposure to Authoritarian Values Leads to Lower Positive Affect, Higher Negative Affect, and Higher Meaning in Life

PLOS ONE

Dear Dr. Womick,

I have read your revised manuscript (PONE-D-21-00255R1) "Exposure to Authoritarian Values Leads to Lower Positive Affect, Higher Negative Affect, and Higher Meaning in Life ".  I appreciate your responsiveness to the comments of the reviewers and myself on the earlier version of this manuscript.  There are, however, a few additional tweaks that need to be made before your manuscript can be accepted for publication.

First, for each study, please indicate when (month and year) the data was collected.

Second, I’m a little confused about your response to reviewer 1’s question about testing the equality of slopes for your ANCOVAs (reviewer 4 had a somewhat related concern).  Note that you don’t actually say ANCOVA in the manuscript, but I’m assuming that’s what you’ve ran.  In your revision, please be more specific about the type of analyses you conducted (for all of your analyses). In response to this reviewer’s comment, you report multiple regression analyses in Table 5 designed to assess the possible moderation of condition on meaning in life by mood.  Although potentially interesting, this doesn’t really address the question of equality of slopes (it hints at it but doesn’t directly address it).  The point is that your main analyses testing for condition on mood controlling for meaning, and condition on meaning controlling for mood, require a test of the covariate by condition interaction to test for slope equality. This is especially important because you rely on means that are adjusted by the covariate(s), and this can be misleading if the slopes are unequal. So, please test and if the interactions are not significant, everything can be reported as is (but please note in your manuscript that this assumption was tested).  If any of the interactions are significant, then the reporting of the adjusted means becomes a little more problematic. There are various options when the interaction is significant, the most common being to test the IV at different levels of the covariate (e.g., the mean and one SD above and one SD below), similar to what you do in the analyses you report in figures 3 - 5.  Another option would be to report only the unadjusted means and corresponding tests. There may be other options as well.

Third, the df in Table 4 seem off (e.g., for Study 5 the df for the test of the adjusted means cannot be correct).  Please verify all values in this (and all other) tables.

That’s it.  Once you submit a revision I’ll review it and make a determination. Note that I won’t be sending this out again for review.

We look forward to receiving your revised manuscript.

Kind regards,

Thomas Holtgraves, Ph.D.

Academic Editor

PLOS ONE
---

## [Author Response · Author response to Decision Letter 1]

3 Aug 2021

Dear Dr. Holtgraves,

Thank you for again providing thoughtful and constructive comments on our manuscript, PONE-D-21-00255R1) "Exposure to Authoritarian Values Leads to Lower Positive Affect, Higher Negative Affect, and Higher Meaning in Life." We have completed revision in response to each of your comments, which we believe continue to improve the quality of our work. Below we detail each of these revisions: 

1. First, for each study, please indicate when (month and year) the data was collected.

Response: These have been added for each Study. 

2. Second, I’m a little confused about your response to reviewer 1’s question about testing the equality of slopes for your ANCOVAs (reviewer 4 had a somewhat related concern). Note that you don’t actually say ANCOVA in the manuscript, but I’m assuming that’s what you’ve ran. In your revision, please be more specific about the type of analyses you conducted (for all of your analyses). In response to this reviewer’s comment, you report multiple regression analyses in Table 5 designed to assess the possible moderation of condition on meaning in life by mood. Although potentially interesting, this doesn’t really address the question of equality of slopes (it hints at it but doesn’t directly address it). The point is that your main analyses testing for condition on mood controlling for meaning, and condition on meaning controlling for mood, require a test of the covariate by condition interaction to test for slope equality. This is especially important because you rely on means that are adjusted by the covariate(s), and this can be misleading if the slopes are unequal. So, please test and if the interactions are not significant, everything can be reported as is (but please note in your manuscript that this assumption was tested). If any of the interactions are significant, then the reporting of the adjusted means becomes a little more problematic. There are various options when the interaction is significant, the most common being to test the IV at different levels of the covariate (e.g., the mean and one SD above and one SD below), similar to what you do in the analyses you report in figures 3 - 5. Another option would be to report only the unadjusted means and corresponding tests. There may be other options as well.

Response: We ran moderated regression models to test for interactions of these variables with condition to address whether slopes for mood differed across conditions. Z-scores are provided testing for these differences. These were already reported for Study 2 on p. 23 of the Supplement. We have added additional information for Studies 3-5 on p. 30 of the main text. These results show that in no case did the slopes in the authoritarian vs. egalitarian conditions significantly differ. In Study 3 (only), NA was more strongly negatively related to meaning in life in the control condition than in the authoritarian condition, z = 2.21, p = .027. We do not view this result as particularly informative. Note as well that it occurred in a direction opposite of the notion that enhanced anger in response to the authoritarian condition boosted meaning in life. Considering results from all of the studies together, these data suggest that the relationship between mood and meaning in life did not vary across conditions. Additionally, throughout the manuscript we have explicitly stated each type of analysis conducted. 

3. Third, the df in Table 4 seem off (e.g., for Study 5 the df for the test of the adjusted means cannot be correct). Please verify all values in this (and all other) tables.

Response: Thank you for your close attention to our work and for catching this error. We have corrected the values in Table 4, and verified the other values are correct. 

Thank you again for the continued opportunity for our work to be considered for publication. 

Sincerely,

Jake Womick

---

## [Editor Report · Decision Letter 2]

16 Aug 2021

Exposure to Authoritarian Values Leads to Lower Positive Affect, Higher Negative Affect, and Higher Meaning in Life

PONE-D-21-00255R2

Dear Dr. Womick,

We’re pleased to inform you that your manuscript has been judged scientifically suitable for publication and will be formally accepted for publication once it meets all outstanding technical requirements.

Kind regards,

Thomas Holtgraves, Ph.D.

Academic Editor

PLOS ONE

---

## [Editor Report · Acceptance letter]

20 Aug 2021

PONE-D-21-00255R2 

Exposure to Authoritarian Values Leads to Lower Positive Affect, Higher Negative Affect, and Higher Meaning in Life 

Dear Dr. Womick:

I'm pleased to inform you that your manuscript has been deemed suitable for publication in PLOS ONE. Congratulations! Your manuscript is now with our production department. 

Kind regards, 

on behalf of

Dr. Thomas Holtgraves 

Academic Editor

PLOS ONE